# Low-Salt Diet Regulates the Metabolic and Signal Transduction Genomic Fabrics, and Remodels the Cardiac Normal and Chronic Pathological Pathways

**Dumitru A. Iacobas** [1,*], **Haile Allen** [1] **and Sanda Iacobas** [2]

[1] Undergraduate Medical Academy, Prairie View A&M University, Prairie View, TX 77446, USA; haileallen0829@gmail.com

[2] Department of Pathology, New York Medical College, Valhalla, NY 10595, USA; sandaiacobas@gmail.com

* Correspondence: daiacobas@pvamu.edu

**Abstract:** Low-salt diet (LSD) is a constant recommendation to hypertensive patients, but the genomic mechanisms through which it improves cardiac pathophysiology are still not fully understood. Our publicly accessible transcriptomic dataset of the left ventricle myocardium of adult male mice subjected to prolonged LSD or normal diet was analyzed from the perspective of the Genomic Fabric Paradigm. We found that LSD shifted the metabolic priorities by increasing the transcription control for fatty acids biosynthesis while decreasing it for steroid hormone biosynthesis. Moreover, LSD remodeled pathways responsible for cardiac muscle contraction (CMC), chronic Chagas (CHA), diabetic (DIA), dilated (DIL), and hypertrophic (HCM) cardiomyopathies, and their interplays with the glycolysis/glucogenesis (GLY), oxidative phosphorylation (OXP), and adrenergic signaling in cardiomyocytes (ASC). For instance, the statistically ($p < 0.05$) significant coupling between GLY and ASC was reduced by LSD from 13.82% to 2.91% (i.e., $-4.75\times$), and that of ASC with HCM from 10.50% to 2.83% ($-3.71\times$). The substantial up-regulation of the CMC, ASC, and OXP genes, and the significant weakening of the synchronization of the expression of the HCM, CHA, DIA, and DIL genes within their respective fabrics justify the benefits of the LSD recommendation.

**Keywords:** adrenergic signaling in cardiomyocytes; cardiac muscle contraction; Chagas disease; diabetic cardiomyopathy; dilated cardiomyopathy; glycerolipid metabolism; glycolysis/glucogenesis; hypertrophic cardiomyopathy; purine metabolism; steroid hormone biosynthesis

## 1. Introduction

The role of excessive salt intake in hypertension and the health benefits of salt reduction are very well documented [1–3]. Although sodium is essential for almost all physiological functions, from nutrient absorption to nervous impulse transmission and muscle contraction [4–6], in excess it adversely impacts the metabolism [7], immunity [8], fibrosis [9], and cardiopulmonary work [10–12] among many other effects. In a rat model, salt-elevated food with NaCl concentration exceeding 4% (like in the human-used processed meats and soups) was shown to exacerbate the development of various types of cardiomyopathy [13] leading to heart failure.

Careful gene expression studies related high salt consumption to transcriptomic alterations in the cardiac tissue and the occurrence of cardiovascular diseases [14,15]. It was reported that excessive salt specifically enriched the pathways of hypertrophic cardiomyopathy (HCM) in the male mouse, and that of dilated cardiomyopathy (DIL) in the female mouse [16]. However, hyponatremia, defined as a serum sodium of <135 mmol/L, is an independent risk factor for higher morbidity and mortality rates [17].

Nevertheless, all previous transcriptomic studies were limited to identifying the up- and down-regulated genes and what functional pathways have been enriched in response to a specific salt diet. As shown in this report, the expression levels of the genes represent a

tiny percentage of the information that can be taken from high-throughput gene expression NG RNA-sequencing and microarray platforms.

The (Cardio)Genomic Fabric Paradigm (GFP, [18]) approach makes the most theoretically possible from quantifying expressions of thousands of genes at a time on several biological replicas. In addition to the average expression level, GFP also takes into account the variations in transcript abundances across biological replicas and the degree of expression correlations of all gene pairs.

Here, we analyze how reducing the salt intake affects the left ventricle metabolic pathways and the functional pathways of cardiac muscle contraction (CMC) and those of Chagas (CHA) [19,20], diabetic (DIA) [21,22], DIL [23], and HCM [24,25] cardiomyopathies. The genes involved in the analyzed pathways were selected using the Kyoto Encyclopedia of Genes and Genomes (KEGG) [26].

## 2. Materials and Methods

### 2.1. Experimental Data

We analyzed the gene expression data from a previous Agilent microarray experiment that profiled the transcriptomes of the left heart ventricle myocardia of 16 weeks old C57Bl/6j male mice subjected for the last 8 weeks of their lives to normal ("N", 0.4% Na) or low-("L", 0.05% Na) salt diet. Four male mice from the same litter were used for each of the two conditions to minimize the biological variability. Any microarray spot with corrupted pixels or with the foreground fluorescence less than twice the background fluorescence in one condition was eliminated from the analysis. The experimental protocol and raw and normalized expression data are publicly accessible in the Gene Expression Omnibus (GEO) of the (USA) National Center for Biotechnology Information (NCBI) [27].

### 2.2. Primary Independent Characteristics of Individual Genes and Functional Pathways

Every quantified gene from normal (N) or low-salt (L) diet-fed animals was characterized by three independent measures deduced from the raw microarray data using the algorithms presented in Appendix A. These primary measures are: average expression level (AVE, definition A1), relative expression variation (REV, definition A3), and expression correlation (*COR*, definition A5), with each other genes in the same condition. Each primary characteristic of individual genes were also averaged over the genes included in specific functional pathway (definitions: A2, A4, A6).

One can attach statistical significance to the expression coordination of two genes. Thus, the $p < 0.05$ statistically significant correlations between genes probed by single microarray spots are when:

a. $COR_{i,j}^{(c)} \geq 0.951 \rightarrow$ genes $i$ and $j$ are synergistically expressed $\rightarrow$ their expression levels oscillate in phase across biological replicas (i.e., simultaneously going up or down);

b. $COR_{i,j}^{(c)} \leq -0.951 \rightarrow$ genes $i$ and $j$ are antagonistically expressed $\rightarrow$ their expression levels oscillate in antiphase across biological replicas (i.e., when one goes up, the other goes down—and when one goes down, the other goes up);

c. $\left| COR_{i,j}^{(c)} \right| < 0.05 \rightarrow$ genes $i$ and $j$ are independently expressed $\rightarrow$ there is no correlation between their expression oscillations.

When both paired genes were probed by two microarray spots, the cut-off for statistically significant synergistic/antagonistic correlation becomes $\left| COR_{i,j}^{(c)} \right| \geq 0.71$; for three spots it is $\left| COR_{i,j}^{(c)} \right| \geq 0.58$ and so on; the cut-off for $p < 0.05$ statistical significance decreases when the number of probing spots increases. One can get the cut-off values from the available online calculator [28]).

### 2.3. Derived Characteristics of Individual Genes

The above primary characteristics of individual genes can be reworked as presented in Appendix to define the useful composite quantifiers: relative expression control (REC,

definition A7), coordination degree (COORD, definition A9), and gene commanding height (GCH, definition A12). REC is proportional to the strength of the cellular homeostatic mechanisms that control the transcript abundance, limiting the expression fluctuations caused by the stochastic nature of the transcription chemical reactions. COORD indicates how influential that gene is for the expression of all other genes. Finally, GCH is used to establish the gene hierarchy, the top gene (largest GCH) being the Gene Master Regulator of that phenotype [29], the best target for personalized gene therapy [30].

All derived characteristics of individual genes were also averaged over selected KEGG-constructed functional pathways (definitions: A8, A10, A11, A13).

### 2.4. Quantification of Transcriptomic Changes

2.4.1. Significant Regulation of the Average Expression Value

A gene was considered as significantly regulated by the low-salt diet if its expression ratio x (negative for down-regulation) satisfied an absolute fold-change condition and the *p*-value *p* of the heteroscedastic *t*-test of the equality of the two average expressions was less than 0.05. Any uniform cut-off for the absolute fold-change (such as $1.5\times$ or $2.0\times$) might be too stringent for stably expressed genes and low technical noise of the probing microarray spots, or too lax for highly variably expressed genes and high technical noise. Therefore, we use it to calculate the absolute fold-change cut-off "CUT" for every single transcript from the corresponding REVs in the compared conditions (Inequalities A14 in Appendix).

2.4.2. Weighted Individual (Gene) Regulation (WIR) and Weighted Pathway Regulation (WPR)

Presenting the transcriptomic changes as percentages of statistically significant up-/down-regulated out of quantified genes means implicitly considering that only these genes modified the transcriptome, and their contributions were Uniform $+1/-1$. A better indicator would be the expression ratio "x" (negative for down-regulation), the algebraic form of the absolute fold-change "$|x|$". Instead, we consider the weighted individual (gene) regulation (WIR) that is applied to any gene regardless of its regulation status. WIR weights the gene contribution to the overall expression regulation through the net fold-change ($|x|-1$) and the confidence (1-*p*-value) of the regulation (Formula (A15)).

The weighted pathway regulation (WPR) is the square root of the average $(WIR)^2$ over the genes associated with that functional pathway (Formula (A16)).

2.4.3. Regulation of the Expression Control and Expression Coordination

Regulation of the expression control of individual genes and a pathway were computed according to the Formulas (A17) and (A18) and that of the expression coordination of individual genes with Formula (A19). Regulation of the coordination degree within a functional pathway and between two pathways were computed according to the Formulas (A20) and (A21).

### 2.5. Functional Pathways

We analyzed the effects of the low-salt diet on the following KEGG-constructed metabolic functional pathways:

(i)  Carbohydrate metabolism:

- (FRU) mmu00051 Fructose and manose metabolism [31];
- (GAL) mmu00052 Galactose metabolism [32];
- (GLY) mmu00010 Glycolysis/glucogenesis [33];
- (INO) mmu00562 Inositol phosphate metabolism [34].

(ii)  Energy metabolism:

- (OXP) mmu00190 Oxidative phosphorylation [35].

(iii)  Lipid metabolism:

- (FAB) mmu00061 Fatty acid biosynthesis [36],

- (GLM) mmu00561 Glycerolipid metabolism [37],
- (GPL) mmu00564 Glycerophospholipid metabolism [38],
- (STB) mmu00100 Steroid biosynthesis [39],
- (SHB) mmu00140 Steroid hormone biosynthesis [40].

(iv)  Nucleotide metabolism:

- (PUM) mmu00230 Purine metabolism [41],
- (PYR) mmu00240 Pyrimidine metabolism [42].

(v)  Amino acid metabolism:

- (CYS) mmu00270 Cysteine and methionine metabolism [43],
- (GLU) mmu00480 Glutathione metabolism [44],
- (THY) mmu00350 Thyrosine metabolism [45],
- (VLI) Valine, leucine, and isoleucine degradation [46].

(vi)  Glycan biosynthesis and metabolism:

- (NGL) mmu00510 N-Glycan biosynthesis [47].

(vii)  Xenobiotics biodegradation and metabolism:

- (DRC) mmu00982 Drug metabolism—cytochrome P450 [48],
- (DOE) mmu00983 Drug metabolism—other enzymes [49].

A particular interest was given to the modification of the (ASC) mmu04261 adrenergic signaling in cardiomyocytes [50], and (CMC) mmu04260 cardiac muscle contraction [51] circulatory system functional pathways.

We then determined how the reduced salt remodeled the pathways of the (CHA) mmu05142 Chagas disease [52], (DIA) mmu05415 diabetic cardiomyopathy [53], (DIL) mmu05414 dilated cardiomyopathy [54], and (HCM) mmu05410 hypertrophic cardiomyopathy [55] cardiac diseases.

We have also identified the significantly regulated genes in the KEGG-constructed signaling pathways of MAPK (mmu04010 [56]), PIK3-Akt (mmu04151 [57]), Rap1 (mmu04015 [58]), Ras (mmu04014 [59]), Chemokine (mmu04062 [60]), Calcium (mmu04020 [61]), cAMP (mmu04024 [62]), cGMP-PKG (mmu04022 [63]), mTOR (04150 [64]), and Wnt (mmu04150 [65]). Finally, we have also looked for the effects of a low-salt diet on the (CEN), central carbon metabolism in cancer (mmu05230 [66]); and (CHO), choline metabolism in cancer (mmu05231 [67]) pathways.

## 3. Results

### 3.1. The Global Picture

Expressions of 19,605 unigenes were adequately quantified in all four N-samples and four L-samples, many of them averaged over the several microarray spots redundantly probing their transcripts. In addition to the average expression levels across biological replicas (AVE), we computed for every single gene the relative expression variation (REV) and the expression correlation (COR) with each other gene. Thus, by quantifying the expressions of 19,605 genes, we obtained 19,605 AVEs, 19,605 REVs, and $(19,605 \times (19,605 - 1)/2 =) 192,168,210$ CORs, making a total of 192,207,420 values to interpret in each condition and compare between conditions. This total amount of data is 9804 times larger than what would have been used in the traditional analysis limited to AVEs.

As expected, the myofilament genes *Myl3* (myosin, light polypeptide 3; AVE-N = 1134; AVE-L = 1273) and *Actc1* (actin, alpha, cardiac muscle 1; AVE-N = 1105, AVE-L = 987) had the largest (normalized to the median gene) expressions in both normal and low-salt diet. Both *Myl3* and *Actc1* were included by KEGG in the circulatory pathways ASC [50] and CMC [51], and also in HCM [55] and DIL [54] cardiac disease pathways. *Myl3* is a ventricle-specific gene in both adult human [68] and mouse [69] hearts. *Mb* (myoglobin; AVE-N = 1036, AVE-L = 1103), *Slc25a4* (solute carrier family 25 (mitochondrial carrier, adenine nucleotide translocator), member 4; AVE-N = 1011, AVE-L = 984) and *Cox6a2*

(cytochrome c oxidase subunit 6A2; AVE-N = 969, AVE-L = 1012) were also among the top expressed genes in both conditions. Twice the normal levels of *Mb* were recently associated with early acute myocardial infarction [70]; *Slc25a4* is included in the DIA pathway [53] and *Cox6a2* is included in the CMC [51], OXP [35], and DIA pathways.

*Mcph1* (microcephaly, primary autosomal recessive 1; REC-N = 39.05) was the most controlled gene in "N", while *Usp31* (ubiquitin specific peptidase 31, REC-L = 27.93) and *Syt11* (synaptotagmin XI, REC-L = 26.25) were the most controlled genes in "L". *Mcph1* is one determinant of the mitral valve annulus diameter [71], so its high control in the left ventricle myocardium is justified. However, in a low-salt diet, its control is substantially downgraded to REC-L = 2.10, while those of *Usp31* (REC-N = 3.82) and *Syt11* (REC-N = 11.08) were substantially elevated. There is no information to date about the role of Usp31 in cardiac pathophysiology, but Syt11 was reported to decrease the risk of atrial fibrillation [72].

Among all gene pair correlations, we found that the number of ($p < 0.05$) significantly synergistically expressed genes with *Cacna1c* (calcium channel, voltage-dependent, L type, alpha 1C subunit) increased from 260 (/19,604 × 100% = 1.33%) in normal diet to 685 (3.49%) in low-salt diet. The number of significantly antagonistically expressed with *Cacna1c* increased from 398 (2.03%) to 467 (2.38%), and that of the independently expressed increased from 450 (2.29%) to 699 (3.56%). Altogether, the coordination degree of *Cacna1c* with all other ventricular genes increased from 1.07% to 2.31%. *Cacna1c* is an important gene for several signaling pathways (ASC [51], calcium [61], cAMP [62], cGPM-PKG [63], MAPK [56]), all five types of synapses [73], as well as CMC [51], and the cardiomyopathies (DIL [54] and HCM [55]).

### 3.2. Independence of the Three Types of Primary Expression Characteristics of Individual Genes

Figure 1 illustrates the independence of the three primary types of characteristics (AVE, REV, COR) for the 55 quantified GLY genes in the two conditions. We selected the sodium/calcium exchanger Slc8a1 (solute carrier family 8 member A1), involved in several KEGG-constructed signaling pathways (ASC [50], calcium [61], cGMP-PKG) [63], as well as in CMC [51] and the cardiomyopathies DIL [54] and HCM [55], to illustrate the expression correlation.

The independence of these measures is visually evident. Note that there are little differences between the AVE values in the two dietary conditions. In this pathway, only one gene, *Dlat* (dihydrolipoamide S-acetyltransferase (E2 component of pyruvate dehydrogenase complex; x = 1.26, CUT = 1.23) was up-regulated, and two genes, *Aldh3a2* (aldehyde dehydrogenase family 3, subfamily A2; x = −1.46, CUT = 1.20) and *Pck2* (phosphoenolpyruvate carboxykinase 2; x = −2.80, CUT = 2.48), were down-regulated by LSD. However, the differences are moderately larger in the REV values and substantially larger in the COR values. Altogether, these differences indicate that the additional characteristics provide important supplementary descriptors of the transcriptomic changes to which the traditional analysis is blind. For instance, the REV of *Aldh3a2* increased from 1.09% in "N" to 13.96% in "L" (i.e., by 12.75x), and that of *Minpp1* (multiple inositol polyphosphate histidine phosphatase 1) from 2.48% in "N" to 24.50% in "L" (9.88×). The REV of the mitochondrial gene *Pck2* decreased from 101.47% to 26.41% (i.e., −3.84×).

Expression correlation with Slc8a1 of G6pc3 (glucose 6 phosphatase, catalytic, 3) went from −0.83 to +0.82, while that of *Alob* (aldolase B, fructose-bisphosphate) went from +0.34 to −0.98 ($p < 0.05$ significant antagonism). There is no information in PubMed about the particular roles of these two genes (*G6pc3*, *Alob*) in cardiac pathophysiology, so that our results may stimulate future investigations.

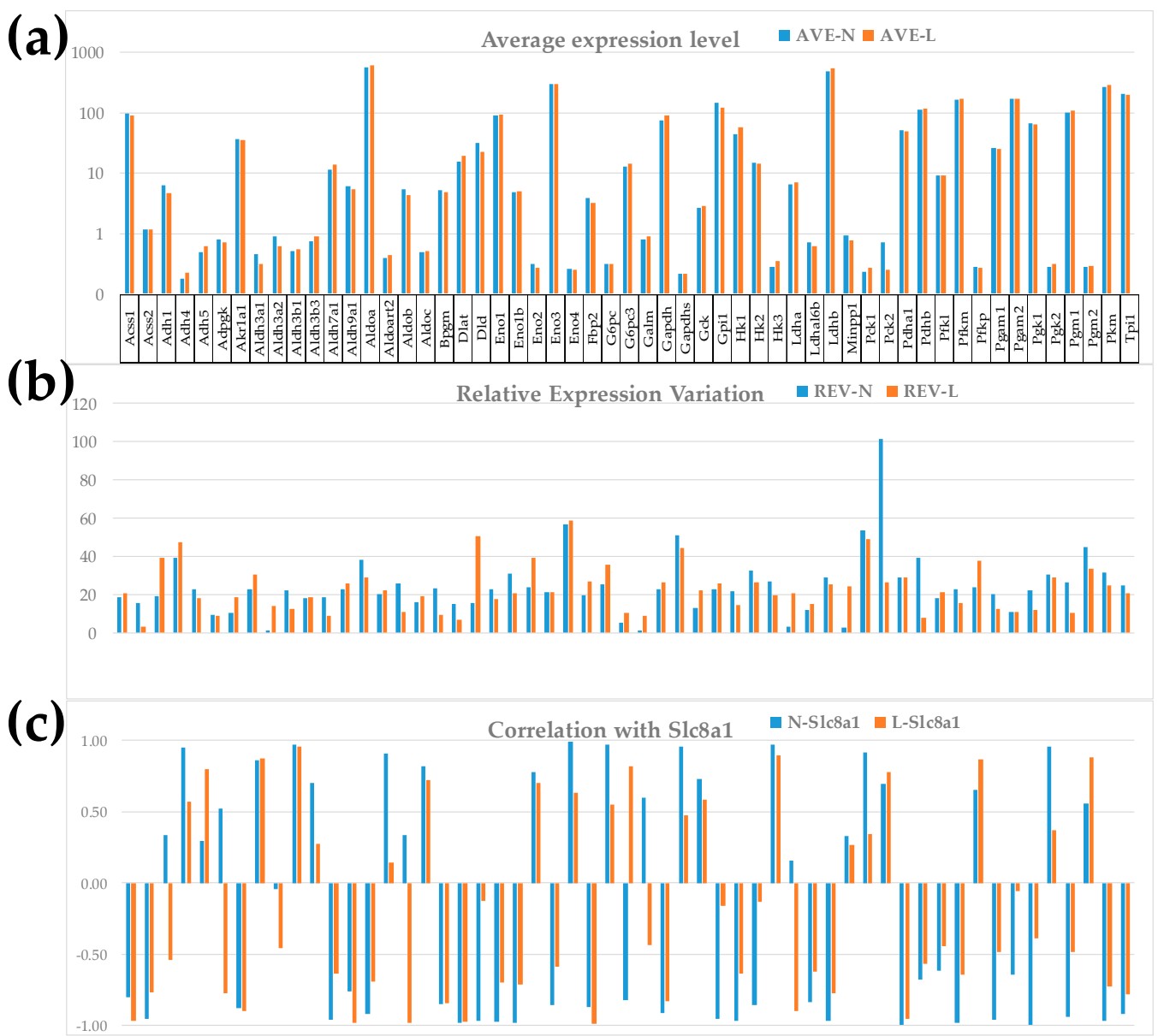

**Figure 1.** The independence of (**a**). AVEs, (**b**). REVs, and (**c**). CORs (with Slc8a1) of the 55 genes quantified within the glycolysis/glucogenesis KEGG-constructed pathway (GLY, [33]). Note the independence of the three characteristics and the changes induced in each of them by the low-salt diet.

### 3.3. Important Derived Characteristics of the Individual Genes

Figure 2 presents the relative expression control, the coordination degree, and the gene commanding height of 55 GLY [33] genes in the two dietary conditions.

The analyses of the derived characteristics unveiled additional interesting effects of the low-salt diet on the GLY genes. For instance, the downgrade of the expression control of *Aldh3a2* (REC-N = 20.64, REC-L = 1.49) and *Galm* (galactose mutarotase; REC-N = 18.76, REC-L = 2.42) led to a substantial reduction in the average REC for this pathway from 2.05 to 1.27. The overall reduction in the expression control of GLY genes in the low-salt condition allows more flexibility in the carbohydrate metabolism.

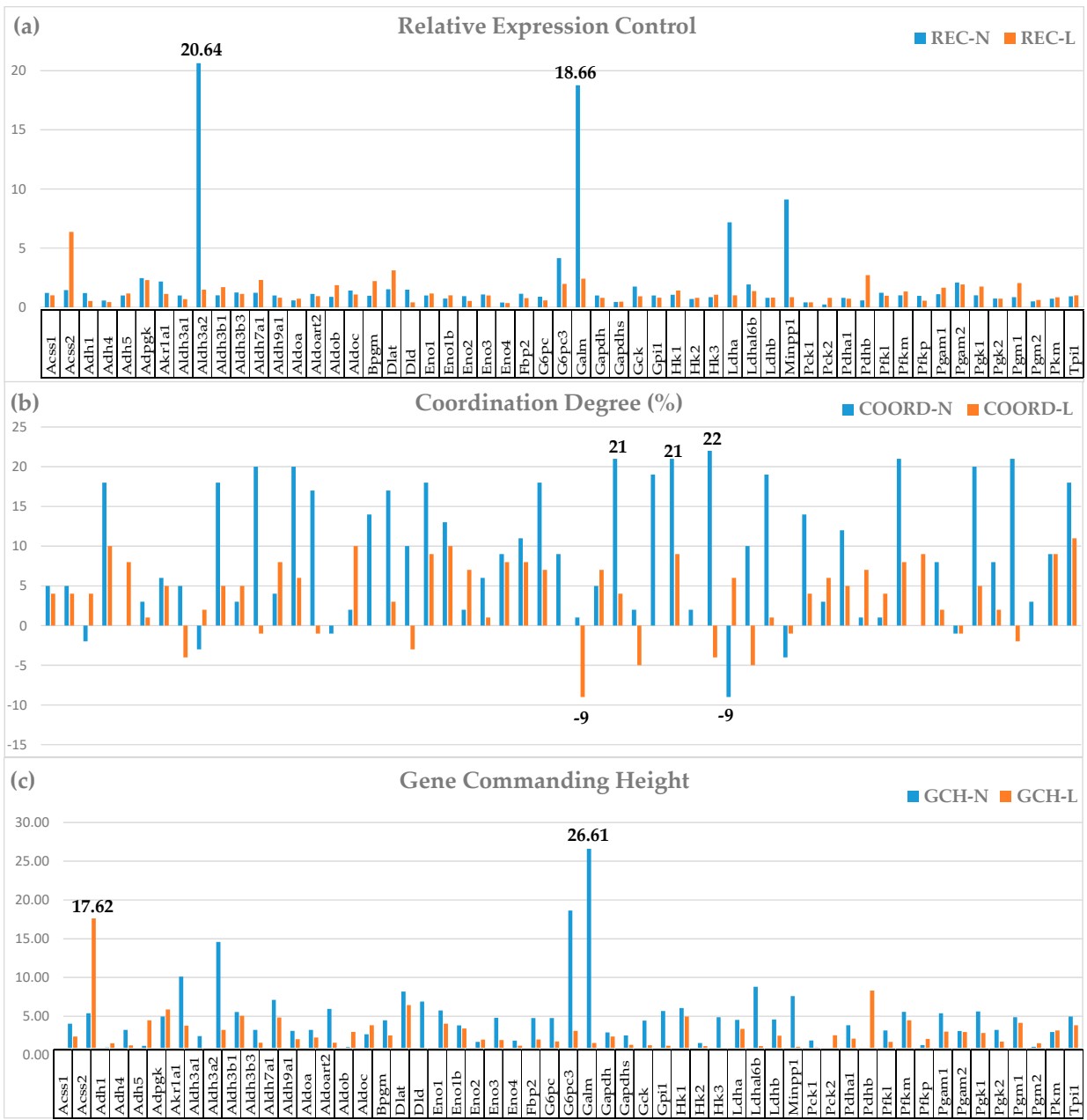

**Figure 2.** Derived characteristics of 55 genes involved in the glycolysis/glucogenesis KEGG-constructed pathway [33]: (**a**). Relative expression control (REC), (**b**). Coordination degree (COORD), (**c**). Gene commanding height (GCH). Note the changes induced by the low-salt diet.

The substantial overall reduction in the coordination degree (from Average COORD-N = 8.98% to Average COORD-L = 3.42%), indicating desynchronization of the genes expressed in this pathway. The most affected genes were *Hk3* (hexokinase 3; COORD-N = 22, COORD-L = −4); *Aldh7a1* (aldehyde dehydrogenase family 7, member A1; COORD-N = 20, COORD-L = −1); *Pgm1* (phosphoglucomutase 1; COORD-N = 21, COORD-L = −2); and *Gapdhs* (glyceraldehyde-3-phosphate dehydrogenase, spermatogenic; COORD-N = 21, COORD-L = 4).

The GCH analysis points to the gene hierarchy change when the salt intake is reduced; genes like *Galm* (GCH-N = 26.61, GCH-L = 1.55) and *Cox4i2* (GCH-N = 33.64, GCH-L = 2.67) become irrelevant in "L".

Owing to the physiological importance, Figure S1 from the Supplementary Materials presents the GCH scores for several genes involved in the KEGG-constructed cardiac

muscle contraction (CMC) pathway [51]. Of note is the substantial downgrade of *Cox4i2* (cytochrome c oxidase subunit 4I2; GCH-N = 33.64, GCH-L = 2.67), a gene also involved in the OXP [35] and DIA [53] pathways. Although none of the mitochondrial cytochrome c oxidase complex genes (*Cox4i1*, *Cox4i2*, *Cox5b*, *Cox6a1*, *Cox6a2*, *Cox6b1*, *Cox6c*, *Cox7a1*, *Cox7a2*, *Cox7a21*, *Cox7b*, *Cox7b2*, *Cox7c*, *Cox8a*, *Cox8b*) was significantly regulated, their average importance (measured by the GCH scores) for the cardiac muscle contraction was downgraded from 7.09 to 2.43. We interpret this result as increased energetic efficiency of the cardiac muscle in the low-salt diet.

### 3.4. Measures of Transcriptomic Regulation

Figure 3 compares the regulation of 50 randomly selected out of the 114 quantified genes included in the purine metabolism KEGG-constructed pathway [41] from the perspective of the Uniform $+1/-1$ contributions, weighted individual regulation (WIR), regulation of expression control ($\Delta$REC), and regulation of the coordination degree ($\Delta$COORD). Nonetheless, the Uniform contribution (the basis of the very popular percentage of up-/down-regulated genes) is limited to the significantly regulated genes and either arbitrarily introduced (e.g., $1.5\times$) or computed for each gene absolute fold-change cut-off.

In contrast, WIR (negative for down-regulation) takes into account all genes. WIR quantifies the total contribution of each gene to the overall transcriptomic alteration that is proportional to the control (here in normal diet) expression level of that gene and its expression ratio (negative for down-regulation) in the experimental condition (low-salt). For instance, while both *Adcy4* (adenylate cyclase 4) and *Prune1* (prune exopolyphosphatase) are significantly down-regulated, (i.e., $-1$ equal contributions to the percentage of the significantly (down-) regulated genes), their WIR measures are substantially different: $\text{WIR}_{\text{Adcy4}} = -3.36$ and $\text{WIR}_{\text{Prune1}} = -48.18$. Likewise, both *Adcy5* (adenylate cyclase 5) and *Adssl1* (adenylosuccinate synthetase like 1) are significantly up-regulated, but with $\text{WIR}_{\text{Adssl1}} = 22.20$, *Adssl1* tops *Adcy5* ($\text{WIR}_{\text{Adcy5}} = 0.13$). The differences came from their dissimilar expression ratios ($x_{\text{Adcy4}} = -1.66$, $x_{\text{Adcy5}} = 1.24$, $x_{\text{Adssl1}} = 1.95$, $x_{\text{Prune1}} = -10.18$) and AVE values ($\text{AVE}_{\text{Adcy4}} = 5.12$, $\text{AVE}_{\text{Adcy5}} = 0.55$, $\text{AVE}_{\text{Adssl1}} = 23.28$, $\text{AVE}_{\text{Prune}} = 6.48$). Thus, beyond the sign (up- or down-), WIR discriminates between the contributions of the regulated genes.

Analysis of the regulation of the expression control produced interesting results for this metabolic pathway, with *Nme1* (NME/NM23 nucleoside diphosphate kinase 1, $\Delta$REC = 370%), and *Adssl1* ($\Delta$REC = 311%) exhibiting the largest increase. *Nme1*, a potential target for metastatic cancer gene therapy [74], was also significantly up-regulated ($x = 1.30$, CUT = 1.26). By contrast, *Gmpr2* (guanosine monophosphate reductase 2, $\Delta$REC = $-153\%$) and *Entpd5* (ectonucleoside triphosphate diphosphohydrolase 5, $\Delta$REC = $-127\%$) presented the largest decrease. Importantly, $\Delta$REC brings non-redundant information about the transcriptomic alteration. Both *Gmpr2* and *Entpd5* were significantly down-regulated by LSD ($x_{\text{Gmpr2}} = -1.37$, $\text{CUT}_{\text{Gmpr2}} = 1.24$; $x_{\text{Entpd5}} = -1.32$, $\text{CUT}_{\text{Entpd5}} = 1.29$).

Analysis of the regulation of the coordination degree revealed substantial decoupling of *Papss2* (3′-phosphoadenosine 5′-phosphosulfate synthase 2; $\Delta$COORD = $-26$) and *Ampd2* (adenosine monophosphate deaminase 2; $\Delta$COORD = $-21$), and increased coupling of *Pde11a* (phosphodiesterase 11A; $\Delta$COORD = 15). While *Pde11a* was also significantly up-regulated ($x = 1.53$) by LSD, *Ampd2* was significantly down-regulated ($x = -1.68$) and expression level of *Papss2* was, practically, not affected ($x = -1.15$).

### 3.5. Correcting the False Hits of the Traditional Significant Regulation Analysis

Overall, we found 1169 (5.96%) unigenes with significant up-regulation and 715 (3.65%) genes with significant down-regulation (the two types satisfying our composite criterion $|x| > \text{CUT} \& p\text{-val} < 0.05$). The flexible cut-off of the absolute fold-change eliminated the false regulated hits ($\text{CUT} > |x| > 1.5 \& p\text{-val} < 0.05$) from the traditional analysis and included the falsely neglected regulated genes ($1.5 > |x| > \text{CUT} \& p\text{-val} < 0.05$). The calculated CUT took values from 1.026 for Syt11 to 3.521 for the purine gene Pde5a

(phosphodiesterase 5A, cGMP-specific). Altogether, our algorithm eliminated 148 falsely considered down-regulated genes and 96 falsely considered up-regulated genes, while adding 685 falsely neglected down-regulated and 553 falsely neglected up-regulated genes.

Table 1 presents examples of falsely considered up-regulated, falsely considered down-regulated, and falsely neglected significantly down- and up-regulated genes. For instance, with x = −2.350, *Ifitm5* (interferon-induced transmembrane protein 5) would have been considered as significantly down-regulated, while it is not, because CUT = 2.427. Likewise, with x = −1.829, the glycerophospholipid metabolism [38] gene *Chkb* (choline kinase beta) would have been considered as significantly down-regulated, while it is not (CUT = 2.633). Similarly, with x = 1.720, the purine/pyrimidine metabolism [41,42] gene *Nt5el* (5′ nucleotidase, ectolike) would have been considered as significantly up-regulated, while it is not, because CUT = 2.153. Another example is *Gclc* (glutamate-cysteine ligase, catalytic subunit), with x = 2.330 and CUT = 2.456. With WIR = 25.41, *Ndufa10* (NADH: ubiquinone oxidoreductase subunit A10), another falsely up-regulated gene (x = 1.505 < CUT = 1.579) had the largest contribution to the overall gene expression change in the low-salt diet. Nonetheless, although not considered by us as significantly regulated, its WIR was included in the WPR of both OXP and DIA functional pathways.

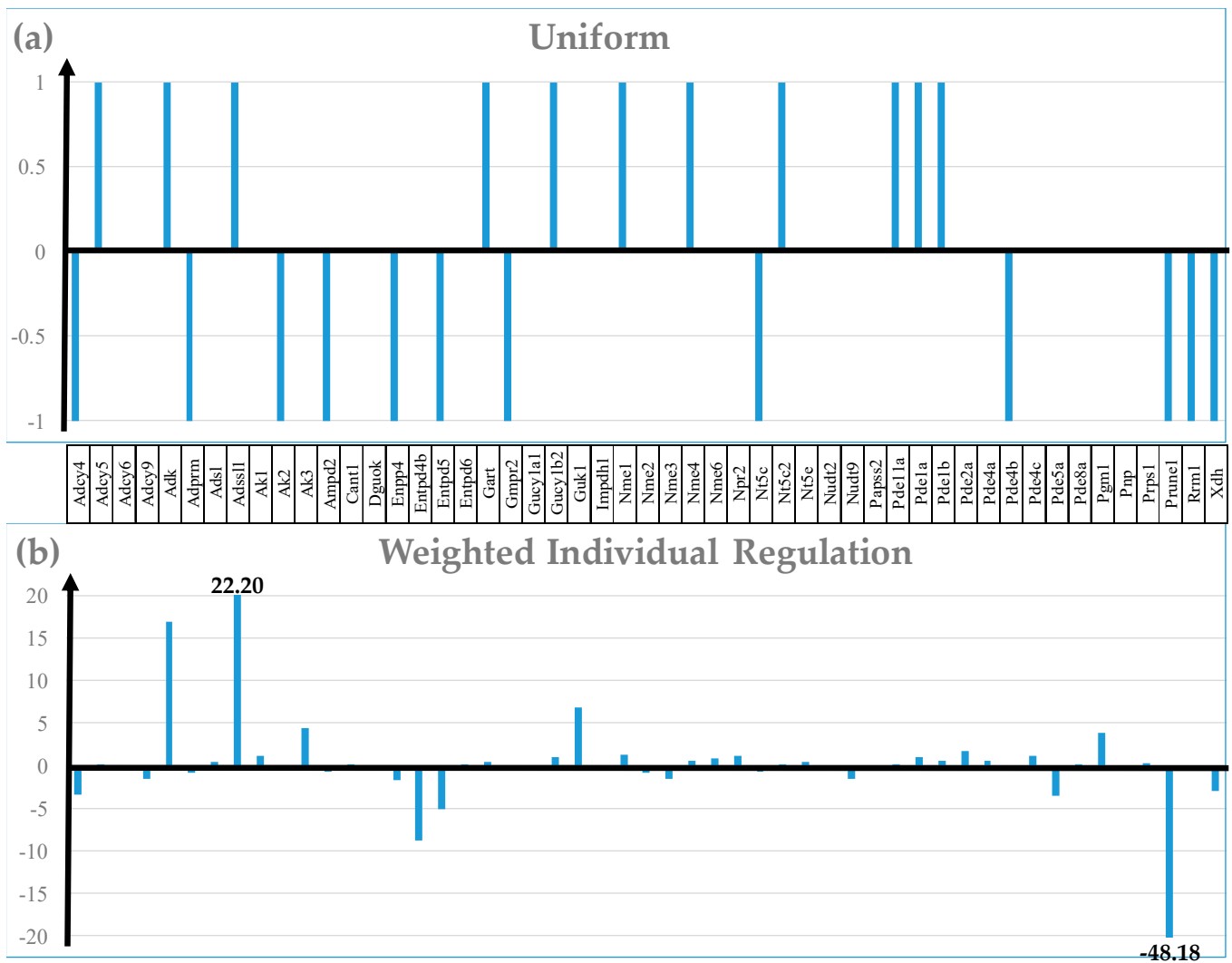

**Figure 3.** *Cont.*

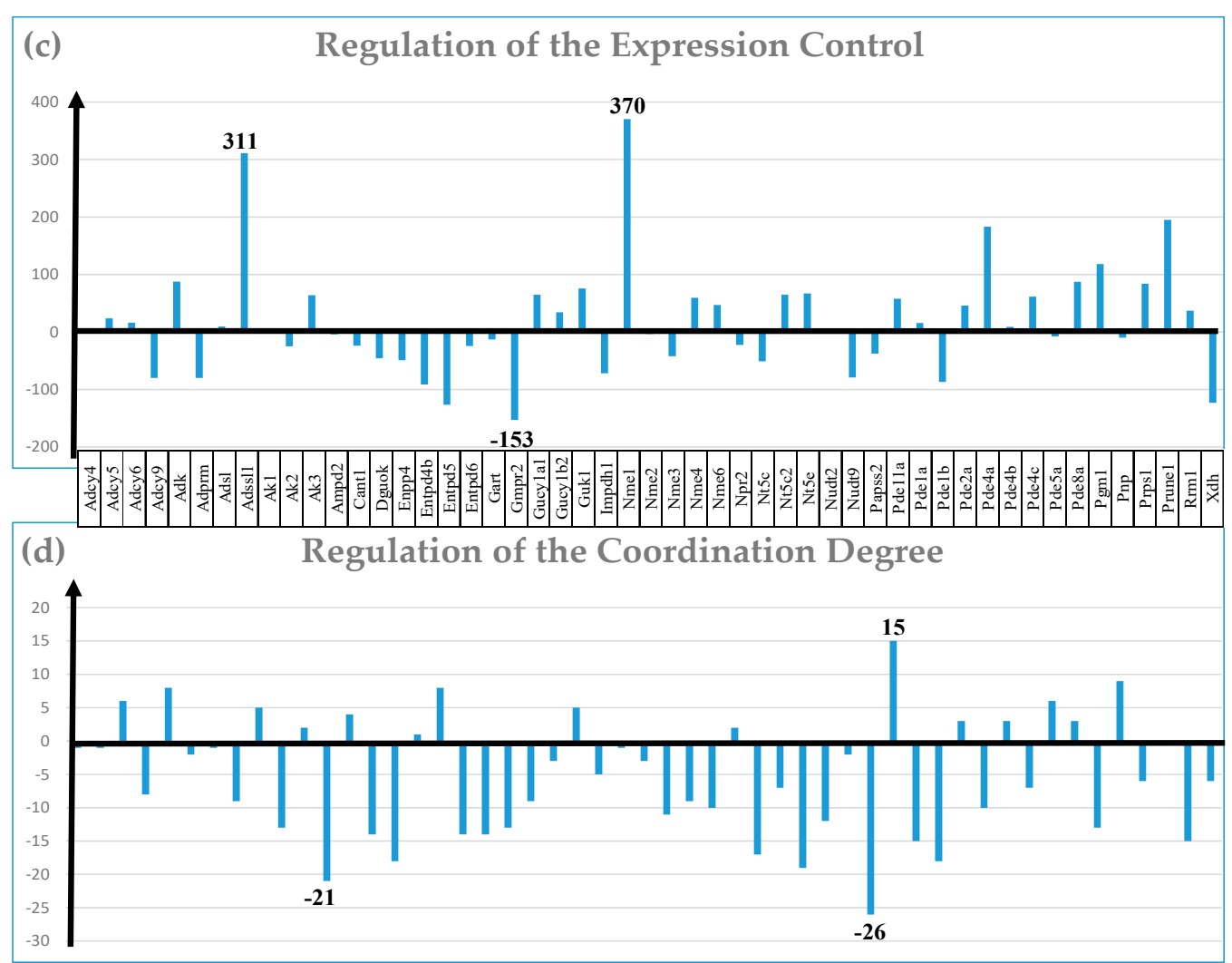

**Figure 3.** Four regulation measures of the transcriptomic characteristics of 50 randomly selected purine metabolism (PUM, [41]) genes: (**a**) Uniform +1/−1 contributions (used to calculate the percentages of up-/down-regulated genes); (**b**) Weighted individual regulation (WIR); (**c**) Regulation of the expression control; (**d**) Regulation of the coordination degree. Note that all measures except Uniform quantify all genes and discriminate their contributions to the overall transcriptomic changes.

**Table 1.** Examples of regulated genes according to the uniform fold-change cut-off = 1.5 that did not pass our |x| > CUT criterion and missed regulated genes in the traditional analysis that satisfied our CUT criterion. All exemplified genes satisfied the *p*-val < 0.05 criterion. X = expression ratio (fold-change, negative for down-regulation), *p* = *p*-value of the heteroscedastic *t*-test of means equality, CUT = absolute fold-change cut-off computed for each gene, WIR = Weighted individual (gene) regulation.

| GENE | DESCRIPTION | X | P | CUT | WIR |
|---|---|---|---|---|---|
| | Falsely down-regulated genes | | | | |
| *Ifitm5* | interferon-induced transmembrane protein 5 | −2.350 | 0.030 | 2.427 | −0.428 |
| *Hinfp* | histone H4 transcription factor | −2.164 | 0.039 | 2.639 | −0.263 |
| *Prdm11* | PR domain containing 11 | −2.000 | 0.026 | 2.170 | −0.376 |
| *Myl7* | myosin, light polypeptide 7, regulatory | −1.887 | 0.022 | 2.468 | −4.566 |
| *Trim71* | tripartite motif-containing 71 | −1.852 | 0.036 | 2.285 | 0.173 |

**Table 1.** *Cont.*

| GENE | DESCRIPTION | X | P | CUT | WIR |
|---|---|---|---|---|---|
| *Usf1* | upstream transcription factor 1 | −1.837 | 0.023 | 1.928 | −0.341 |
| *Chkb* | choline kinase beta | −1.829 | 0.025 | 2.633 | −5.056 |
| *Cntnap5c* | contactin-associated protein-like 5C | −1.824 | 0.025 | 1.922 | −5.270 |
| *Dnajb1* | DnaJ heat shock protein family | −1.812 | 0.034 | 2.129 | −9.529 |
| *Csrnp2* | cysteine-serine-rich nuclear protein 2 | −1.797 | 0.032 | 2.176 | −0.228 |
| | Missed down-regulated genes | | | | |
| *Gsk3b* | glycogen synthase kinase 3 beta | −1.490 | 0.017 | 1.341 | −4.025 |
| *Aldh3a2* | aldehyde dehydrogenase family 3, subfamily A2 | −1.462 | 0.007 | 1.198 | −0.422 |
| *Mapk10* | mitogen-activated protein kinase 10 | −1.455 | 0.028 | 1.306 | −2.712 |
| *Myl2* | myosin, light polypeptide 2, regulatory, cardiac, slow | −1.431 | 0.007 | 1.329 | −0.868 |
| *Tpm2* | tropomyosin 2, beta | −1.421 | 0.027 | 1.359 | −1.751 |
| *Atp5j* | ATP synthase H+ transporting mitochondrial F0 complex subunit F | −1.401 | 0.013 | 1.272 | −0.171 |
| *Gmpr2* | guanosine monophosphate reductase 2 | −1.371 | 0.009 | 1.238 | −0.350 |
| *Enpp4* | ectonucleotide pyrophosphatase/phosphodiesterase 4 | −1.362 | 0.028 | 1.316 | −1.748 |
| *Chat* | choline acetyltransferase | −1.353 | 0.024 | 1.292 | −0.253 |
| *Dbt* | dihydrolipoamide branched chain transacylase E2 | −1.323 | 0.024 | 1.274 | −1.046 |
| | Missed up-regulated genes | | | | |
| *Lpin3* | lipin 3 | 1.372 | 0.004 | 1.146 | 0.366 |
| *Pde1a* | phosphodiesterase 1A, calmodulin-dependent | 1.374 | 0.008 | 1.219 | 0.974 |
| *Gpam* | glycerol-3-phosphate acyltransferase, mitochondrial | 1.374 | 0.019 | 1.214 | 0.427 |
| *B4galt1* | UDP-Gal:betaGlcNAc beta 1,4- galactosyltransferase, polypeptide 1 | 1.391 | 0.005 | 1.334 | 3.184 |
| *Ncf4* | neutrophil cytosolic factor 4 | 1.397 | 0.046 | 1.320 | 0.273 |
| *Bcl2* | B cell leukemia/lymphoma 2 | 1.401 | 0.005 | 1.164 | 0.392 |
| *Ndufc1* | NADH: ubiquinone oxidoreductase subunit C1 | 1.410 | 0.018 | 1.303 | 58.827 |
| *Ikbkg* | inhibitor of kappaB kinase gamma | 1.424 | 0.005 | 1.233 | 0.260 |
| *Atp6v1b2* | ATPase, H+ transporting, lysosomal V1 subunit B2 | 1.438 | 0.045 | 1.381 | 0.265 |
| *Gucy1b2* | guanylate cyclase 1, soluble, beta 2 | 1.490 | 0.034 | 1.426 | 0.943 |
| | Falsely up-regulated genes | | | | |
| *Kif3c* | kinesin family member 3C | 1.706 | 0.009 | 1.832 | 3.179 |
| *Nt5el* | 5′ nucleotidase, ecto-like | 1.720 | 0.028 | 2.153 | 0.097 |
| *Zfp362* | zinc finger protein 362 | 1.758 | 0.024 | 1.852 | 0.637 |
| *Ctsg* | cathepsin G | 1.887 | 0.018 | 1.890 | 0.192 |
| *Tmem231* | transmembrane protein 231 | 1.912 | 0.027 | 2.196 | 0.128 |
| *Adam12* | a disintegrin and metallopeptidase domain 12 | 1.966 | 0.036 | 2.313 | 0.423 |
| *Ftcd* | formiminotransferase cyclodeaminase | 1.979 | 0.033 | 2.214 | 0.163 |
| *Ap1m1* | adaptor-related protein complex AP-1, mu subunit 1 | 2.063 | 0.006 | 2.079 | 11.060 |
| *Lrrc71* | leucine-rich repeat containing 71 | 2.153 | 0.034 | 2.559 | 0.138 |
| *Gclc* | glutamate-cysteine ligase, catalytic subunit | 2.330 | 0.028 | 2.456 | 1.332 |

In contrast, the significant regulation of the diabetic cardiomyopathy [50] gene *Gsk3b* (glycogen synthase kinase 3 beta, x = −1.490, CUT = 1.341) and the purine metabolism [41] gene *Gucy1b2* (guanylate cyclase 1, soluble, beta 2; x = 1.490, CUT = 1.426) would have been neglected. There are other important genes that would have been disconsidered by the traditional 1.5 absolute fold-change cut-off. For instance, with x = −1.178, the Chagas disease [52] gene *Casp8* (Caspase 8) would have been neglected, although it is significantly down-regulated because CUT = 1.159 < |x|. Finally, *Tgfb3* (transforming growth factor, beta 3), included in the functional pathways of the Chagas [52], hypertrophic [55], diabetic [53], and dilated [54] cardiomyopathies, would have also been neglected although CUT = 1.093 < x = 1.166.

Out of the neglected genes, by the traditional analysis, the OXP [35] and DIA [53] gene *Ndufc1* (NADH: ubiquinone oxidoreductase subunit C1) had the largest contribution to the LSD-induced transcriptomic changes from the WIR perspective (WIR = 58.83; x = 1.41 > 1.30 = CUT).

*3.6. Overall Regulation of Expression Level and Transcription Control within Selected Metabolic, Circulatory System, and Cardiac Chronic Diseases' Pathways*

Table 2 presents the percentages of down- and up-regulated out of quantified genes, the weighted pathway regulation (WPR), and the changes in the control of transcript abundances within several selected functional pathways. Unfortunately, not all genes assigned to the respective functional pathways were quantified, either because of not being expressed in the left ventricle, missing the probing spots in the microarrays, or being probed by spots with corrupted pixels during hybridization. For instance, out of 156 genes assigned to ASC by KEGG, we quantified only 130 (i.e., 83.33%), still enough to have a statistically relevant evaluation of the transcriptomic change in this pathway.

From the WPR perspective, the most affected pathways were CMC (WPR = 45.30) and OXP (WPR = 37.42), indicating the major effects of reduced salt on ventricle contraction and energy metabolism. Control of transcript abundances was substantially diminished for steroid hormone biosynthesis, but strengthened for biosyntheses of fatty acids and N-glycan, as well as for oxidative phosphorylation, indicating significant shifts in the cardiomyocyte homeostasis priorities.

**Table 2.** Transcriptomic changes in the studied KEGG-constructed functional pathways. GENES (e.g.,130/156) genes quantified/genes in the pathway, D% = percent down-regulated out of quantified genes, U% = percent up-regulated out of quantified genes, WPR = weighted pathway regulation, ΔREC (%) = percent change in the overall control of transcript abundance in the pathway (negative for reduced control, i.e., increased expression variation).

| mmu | PATH | Description | GENES | D% | U% | WPR | ΔREC (%) |
|---|---|---|---|---|---|---|---|
| 04261 | ASC | Adrenergic signaling in cardiomyocytes | 130/156 | 6.15 | 13.08 | 19.97 | −3.71 |
| 04260 | CMC | Cardiac muscle contraction | 75/87 | 5.33 | 10.67 | 45.30 | −1.38 |
| 05142 | CHA | Chagas disease | 85/103 | 3.61 | 12.05 | 3.31 | −6.71 |
| 05415 | DIA | Diabetic cardiomyopathy | 184/211 | 3.80 | 7.07 | 29.55 | 0.40 |
| 05414 | DIL | Dilated cardiomyopathy | 81/94 | 6.17 | 12.35 | 7.05 | −0.45 |
| 00061 | FAB | Fatty acids biosynthesis | 18/19 | 0.00 | 5.56 | 2.49 | 17.10 |
| 00561 | GLM | Glycerolipid metabolism | 52/63 | 3.85 | 15.38 | 4.63 | −5.88 |
| 00564 | GPL | Glycerophospholipid metabolism | 83/98 | 4.82 | 9.64 | 1.54 | 2.27 |
| 00010 | GLY | Glycolysis/glucogenesis | 55/64 | 3.64 | 1.82 | 5.51 | 6.18 |
| 05410 | HCM | Hypertrophic cardiomyopathy | 78/91 | 6.41 | 8.97 | 6.73 | 2.23 |
| 00510 | NGL | N-Glycan biosynthesis | 50/53 | 4.00 | 4.00 | 14.18 | 14.63 |
| 00190 | OXP | Oxidative phosphorylation | 110/135 | 1.82 | 6.36 | 37.42 | 12.39 |

**Table 2.** *Cont.*

| mmu | PATH | Description | GENES | D% | U% | WPR | ΔREC (%) |
|---|---|---|---|---|---|---|---|
| 00230 | PUM | Purine metabolism | 114/134 | 10.53 | 11.40 | 5.42 | 4.19 |
| 00240 | PYR | Pyrimidine metabolism | 47/56 | 8.51 | 10.64 | 1.64 | −5.83 |
| 00100 | STB | Steroid biosynthesis | 17/20 | 0.00 | 5.88 | 0.69 | −11.37 |
| 00140 | SHB | Steroid hormone biosynthesis | 42/93 | 7.14 | 9.52 | 8.27 | −18.74 |
| 00280 | VLI | Valine, leucine, and isoleucine degradation | 48/57 | 6.25 | 2.08 | 9.28 | 5.72 |
| | ALL | All quantified genes | 19,605 | 3.65 | 5.96 | 15.67 | 0.30 |

### 3.7. Regulated Genes within Selected Metabolic Pathways

Out of the 1169 significantly up-regulated genes, 97 were included in KEGG-constructed metabolic pathways, while within the 715 down-regulated genes, 66 were responsible for metabolism pathways.

Table 3 presents the statistically significantly down- and up-regulated genes in the most affected (as a number of regulated genes) KEGG-constructed metabolic pathways. Importantly, the reduced salt increased several metabolic pathways (more up-regulated than down-regulated genes), including those of the glycerophospholipid, glutathione, and glycerolipid, as well as the oxidative phosphorylation. Notably, we found no significantly down-regulated genes in either the Galactose metabolism or the Tyrosine metabolism.

**Table 3.** Significantly down- (D) and up (U, bold symbols)-regulated genes identified with our CUT-based algorithm from the most affected KEGG-constructed metabolic pathways. Note that the pathways are not mutually exclusive, but partially overlapping. For instance, "Choline metabolism in cancer" and "Central carbon metabolism in cancer" share the genes *Akt1*, *Akt3*, *Egfr*, *Hif1a*, *Kras*, *Mapk1*, *Pdgfra*, and *Pdgfrb*.

| PATHWAY | R | GENES |
|---|---|---|
| Purine metabolism | D | *Adcy4*; *Adprm*; *Ak2*; *Ampd2*; *Enpp4*; *Entpd5*; *Gmpr2*; *Nt5c*; *Pde4b*; *Prune1*; *Rrm1*; *Xdh* |
| | U | *Adcy1*; *Adcy5*; *Adk*; *Adssl1*; *Gart*; *Gucy1b2*; *Nme1*; *Nme4*; *Nt5c2*; *Pde11a*; *Pde1a*; *Pde1b*; *Prps2* |
| Choline metabolism in cancer | D | *Akt3*; *Gpcpd1*; *Mapk10*; *Pdgfd*; *Pdgfra*; *Pdgfrb*; *Rac2* |
| | U | *Akt1*; *Egfr*; *Hif1a*; *Kras*; *Mapk1*; *Pdpk1*; *Pip5k1a*; *Plpp1*; *Plpp2*; *Plpp3*; *Prkca*; *Prkcb*; *Rac1*; *Slc44a1* |
| Drug metabolism—other enzymes | D | *Ces1d*; *Gsta3*; *Gstt1*; *Gstt2*; *Rrm1*; *Xdh* |
| | U | *Cmpk1*; *Gsta4*; *Gstm1*; *Gstm6*; *Gstm7*; *Gstp1*; *Gusb*; *Nat2*; *Nme1*; *Nme4*; *Upp1* |
| Glycerophospholipid metabolism | D | *Adprm*; *Chat*; *Gpcpd1*; *Selenoi* |
| | U | *Etnk2*; *Gpam*; *Lpin3*; *Mboat1*; *Pla1a*; *Plpp1*; *Plpp2*; *Plpp3* |
| Glutathione metabolism | D | *Gsta3*; *Gstt1*; *Gstt2*; *Rrm1* |
| | U | *Chac1*; *Gsta4*; *Gstm1*; *Gstm6*; *Gstm7*; *Gstp1*; *Odc1*; *Srm* |
| Central carbon metabolism in cancer | D | *Akt3*; *Fgfr3*; *Pdgfra*; *Pdgfrb*; *Slc1a5* |
| | U | *Akt1*; *Egfr*; *Hif1a*; *Kras*; *Mapk1*; *Sco2* |
| Drug metabolism—cytochrome P450 | D | *Fmo1*; *Gsta3*; *Gstt1*; *Gstt2* |
| | U | *Fmo5*; *Gsta4*; *Gstm1*; *Gstm6*; *Gstm7*; *Gstp1* |
| Glycerolipid metabolism | D | *Aldh3a2*; *Mgll* |
| | U | *Akr1b8*; *Aldh1b1*; *Gpam*; *Lpin3*; *Mboat1*; *Plpp1*; *Plpp2*; *Plpp3* |
| Pyrimidine metabolism | D | *Cmpk2*; *Entpd5*; *Nt5c*; *Rrm1* |
| | U | *Cmpk1*; *Nme1*; *Nme4*; *Nt5c2*; *Upp1* |

**Table 3.** *Cont.*

| PATHWAY | R | GENES |
|---|---|---|
| Cysteine and methionine metabolism | D | *Agxt2*; *Amd2*; *Mpst* |
| | U | *Adi1*; *Apip*; *Mtap*; *Srm*; *Tst* |
| Inositol phosphate metabolism | D | *Inpp1*; *Isyna1* |
| | U | *Pi4k2a*; *Pik3c2b*; *Pip5k1a*; *Plcd3*; *Synj2* |
| Fructose and mannose metabolism | D | *Pfkfb1* |
| | U | *Akr1b8*; *Gmds*; *Khk*; *Pfkfb3*; *Pfkfb4* |
| Galactose metabolism | U | *Akr1b8*; *B4galt1*; *Gaa*; *Ugp2* |
| Tyrosine metabolism | U | *Comt*; *Dct*; *Mif*; *Th* |

### 3.8. Regulation of Selected Signaling Pathways

In total, we found 607 significantly up-regulated and 350 significantly down-regulated genes included in all KEGG-constructed signaling pathways. Figure 4 presents the localization of the regulated genes in the KEGG-constructed ASC (Adrenergic signaling in cardiomyocytes) [50] pathway. Remarkably, 17 (i.e., 13.08%) of the total of 130 quantified genes in the pathway were up-regulated and 8 (6.15%) were down-regulated.

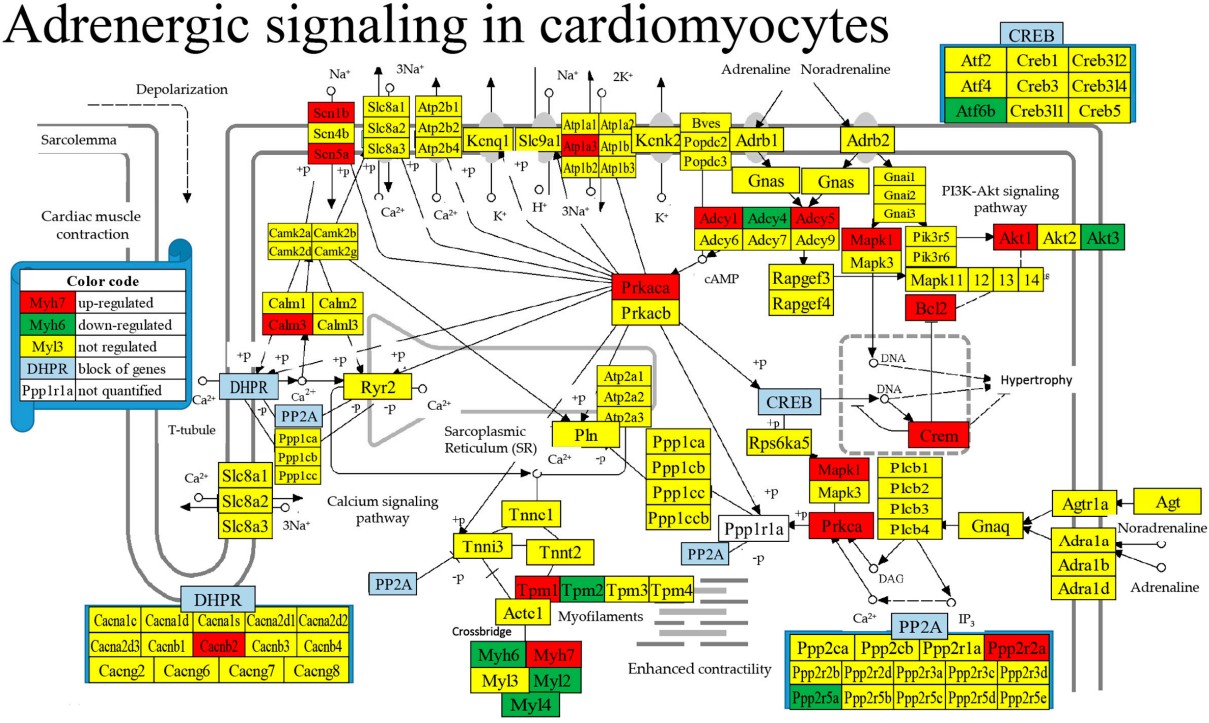

**Figure 4.** Regulated genes in the Adrenergic signaling in cardiomyocyte KEGG-constructed pathway. Owing to space constraints, several genes sharing the same position in the pathway were grouped into blocks of genes presented in panels. Regulated genes: *Adcy1/4/5* (adenylate cyclase 1/4/5), *Akt1/3* (thymoma viral proto-oncogene 1/3), *Atf6b* (activating transcription factor 6 beta), *Atp1a3* (ATPase, Na+/K+ transporting, alpha 3 polypeptides), *Bcl2* (B cell leukemia/lymphoma 2), *Cacnb2* (calcium channel, voltage-dependent, beta 2 subunit), *Calm3* (calmodulin 3), *Crem* (cAMP responsive element modulator), *Fxyd2* (FXYD domain-containing ion transport regulator 2), *Mapk1* (mitogen-activated protein kinase 1), *Myh6/7* (myosin, heavy polypeptide 6, cardiac muscle, alpha/7, cardiac muscle, beta), *Myl2/4* (myosin, light polypeptide 2/4), *Ppp2r2a/5a* (protein phosphatase 2, regulatory subunit B, alpha/regulatory subunit B', alpha), *Prkaca* (protein kinase, cAMP-dependent, catalytic, alpha), *Prkca* (protein kinase C, alpha), *Scn1b* (sodium channel, voltage-gated, type I, beta), *Scn5a* (sodium channel, voltage-gated, type V, alpha), *Tpm1/2* (tropomyosin 1 alpha/2 beta).

The large numbers of regulated genes within the ten signaling pathways from Tables 4 and 5 indicate the high impact of the reduced salt intake diet on heart physiology. Moreover, the 1.73 U/D ratio shows that the diminished sodium increased the overall signaling. Of note is the partial overlap of the pathways; genes such as *Akt1* are listed in all signaling pathways except calcium, and Wnt. With 50 (36U + 14D) and 45 (28I + 17D), respectively, MAPK signaling and PIK3-Akt signaling top the list of the most regulated signaling pathways.

**Table 4.** Up- (U) and down (D)-regulated genes from top five altered KEGG-constructed signaling pathways. Numbers before "U" and "D" indicate how many up-and down-regulated genes were quantified in the respective signaling pathway.

| MAPK | | PI3K-Akt | | Rap1 | | Ras | | Chemokine | |
|---|---|---|---|---|---|---|---|---|---|
| **36U** | **14D** | **28U** | **17D** | **28U** | **13D** | **27U** | **11D** | **21U** | **10D** |
| *Akt1* | *Akt3* | *Akt1* | *Akt3* | *Adcy1* | *Adcy4* | *Abl2* | *Akt3* | *Adcy1* | *Adcy4* |
| *Cacnb2* | *Cacna1g* | *Bcl2* | *Atf6b* | *Adcy5* | *Adora2a* | *Akt1* | *Fgfr3* | *Adcy5* | *Akt3* |
| *Crk* | *Fgfr3* | *Cdkn1a* | *Ddit4* | *Adora2b* | *Akt3* | *Calm3* | *Igf2* | *Akt1* | *Cxcl11* |
| *Csf1* | *Hspa1a* | *Col4a1* | *Epor* | *Akt1* | *Fgfr3* | *Csf1* | *Mapk10* | *Ccl21b* | *Cxcl14* |
| *Dusp6* | *Igf2* | *Col4a2* | *Fgfr3* | *Calm3* | *Map2k6* | *Efna3* | *Pdgfd* | *Ccl6* | *Dock2* |
| *Dusp8* | *Map2k6* | *Col4a5* | *Foxo3* | *Crk* | *P2ry1* | *Egfr* | *Pdgfra* | *Ccr7* | *Foxo3* |
| *Efna3* | *Map3k11* | *Csf1* | *Gsk3b* | *Csf1* | *Pdgfd* | *Ets1* | *Pdgfrb* | *Crk* | *Gsk3b* |
| *Egfr* | *Map3k2* | *Efna3* | *Igf2* | *Efna3* | *Pdgfra* | *Exoc2* | *Rac2* | *Cx3cr1* | *Rac2* |
| *Fgf18* | *Mapk10* | *Egfr* | *Mlst8* | *Egfr* | *Pdgfrb* | *Fgf18* | *Rapgef5* | *Gnb3* | *Rhoa* |
| *Gadd45b* | *Max* | *Eif4e* | *Pck2* | *Enah* | *Prkd2* | *Gnb3* | *Rgl1* | *Gng7* | *Stat2* |
| *Gna12* | *Pdgfd* | *Fgf18* | *Pdgfd* | *Fgf18* | *Rac2* | *Gng7* | *Rhoa* | *Grk3* | |
| *Ikbkg* | *Pdgfra* | *Gnb3* | *Pdgfra* | *Itgal* | *Rapgef5* | *Ikbkg* | | *Ikbkg* | |
| *Irak1* | *Pdgfrb* | *Gng7* | *Pdgfrb* | *Itgb1* | *Rhoa* | *Kras* | | *Kras* | |
| *Kras* | *Rac2* | *Ikbkg* | *Ppp2r5a* | *Itgb2* | | *Mapk1* | | *Mapk1* | |
| *Lamtor3* | | *Il4ra* | *Sgk1* | *Kras* | | *Mras* | | *Prkaca* | |
| *Map3k3* | | *Itga9* | *Thbs2* | *Krit1* | | *Nf1* | | *Prkcb* | |
| *Map3k7* | | *Itgb1* | *Tnxb* | *Mapk1* | | *Ngf* | | *Prkcd* | |
| *Mapk1* | | *Itgb6* | | *Mras* | | *Pla1a* | | *Ptk2b* | |
| *Mapt* | | *Kras* | | *Ngf* | | *Prkaca* | | *Rac1* | |
| *Mknk2* | | *Mapk1* | | *Pard6a* | | *Prkca* | | *Stat5b* | |
| *Mras* | | *Ngf* | | *Pfn1* | | *Prkcb* | | *Tiam1* | |
| *Myd88* | | *Pdpk1* | | *Prkca* | | *Rab5a* | | | |
| *Nf1* | | *Ppp2r2a* | | *Prkcb* | | *Rab5b* | | | |
| *Ngf* | | *Prkca* | | *Rac1* | | *Rac1* | | | |
| *Ppp3ca* | | *Rac1* | | *Rap1gap* | | *Ralgapa2* | | | |
| *Prkaca* | | *Thbs1* | | *Sipa1l2* | | *Stk4* | | | |
| *Prkca* | | *Thbs4* | | *Thbs1* | | *Tiam1* | | | |
| *Prkcb* | | *Tlr2* | | *Tiam1* | | | | | |
| *Ptpn5* | | | | | | | | | |
| *Rac1* | | | | | | | | | |
| *Relb* | | | | | | | | | |

**Table 4.** *Cont.*

| MAPK | | PI3K-Akt | | Rap1 | | Ras | | Chemokine | |
|---|---|---|---|---|---|---|---|---|---|
| **36U** | **14D** | **28U** | **17D** | **28U** | **13D** | **27U** | **11D** | **21U** | **10D** |
| *Srf* | | | | | | | | | |
| *Stk3* | | | | | | | | | |
| *Stk4* | | | | | | | | | |
| *Tgfb3* | | | | | | | | | |
| *Traf2* | | | | | | | | | |

**Table 5.** Up- (U) and down (D)-regulated genes from the calcium, cAMP, cGMP-PKG, mTOR (mammalian (mechanistic) target of rapamycin), and Wnt (wingless-type MMTV integration site family) KEGG-constructed signaling pathways. Numbers before symbols "U" and "D" indicate how many up-and down-regulated genes were quantified in the respective signaling pathway.

| Calcium | | cAMP | | cGMP-PKG | | mTOR | | Wnt | |
|---|---|---|---|---|---|---|---|---|---|
| **15U** | **14D** | **14U** | **11D** | **15U** | **10D** | **16U** | **9D** | **13U** | **12D** |
| *Adcy1* | *Adcy4* | *Adcy1* | *Adcy4* | *Adcy1* | *Adcy4* | *Akt1* | *Akt3* | *Crebbp* | *Fzd4* |
| *Adora2b* | *Adora2a* | *Adcy5* | *Adora2a* | *Adcy5* | *Akt3* | *Atp6v1b2* | *Castor2* | *Csnk2a1* | *Gpc4* |
| *Asph* | *Cacna1g* | *Akt1* | *Akt3* | *Adra2b* | *Atf6b* | *Clip1* | *Ddit4* | *Dvl1* | *Gsk3b* |
| *Calm3* | *Fgfr3* | *Atp1a3* | *Edn1* | *Akt1* | *Itpr2* | *Dvl1* | *Fzd4* | *Map3k7* | *Mapk10* |
| *Egfr* | *Grm1* | *Calm3* | *Mapk10* | *Atp1a3* | *Itpr3* | *Eif4e* | *Gsk3b* | *Notum* | *Porcn* |
| *Fgf18* | *Itpr2* | *Crebbp* | *Myl9* | *Calm3* | *Myh6* | *Kras* | *Mlst8* | *Ppp3ca* | *Prickle1* |
| *Ngf* | *Itpr3* | *Fxyd2* | *Pde4b* | *Fxyd2* | *Myl9* | *Lamtor3* | *Rhoa* | *Prkaca* | *Rac2* |
| *Pde1a* | *Mst1r* | *Hcn2* | *Ppp1r12a* | *Gna12* | *Mylk4* | *Lpin3* | *Rictor* | *Prkca* | *Rhoa* |
| *Pde1b* | *Mylk4* | *Mapk1* | *Ppp1r1b* | *Gtf2ird1* | *Ppp1r12a* | *Mapk1* | *Sgk1* | *Prkcb* | *Sfrp5* |
| *Plcd3* | *P2rx1* | *Prkaca* | *Rac2* | *Gucy1b2* | *Rhoa* | *Pdpk1* | | *Rac1* | *Sox17* |
| *Ppp3ca* | *Pdgfd* | *Rac1* | *Rhoa* | *Mapk1* | | *Prkca* | | *Smad3* | *Tle2* |
| *Prkaca* | *Pdgfra* | *Sst* | | *Myh7* | | *Prkcb* | | *Wnt1* | *Tle3* |
| *Prkca* | *Pdgfrb* | *Sstr5* | | *Nppb* | | *Stradb* | | *Wnt5b* | |
| *Prkcb* | *Phkg1* | *Tiam1* | | *Ppp3ca* | | *Wdr59* | | | |
| *Ptk2b* | | | | *Srf* | | *Wnt1* | | | |
| | | | | | | *Wnt5b* | | | |

### 3.9. Regulated Genes within Pathways of Selected Cardiac Diseases

Figure 5 presents the positions of the 10 (i.e., 12.20%) up-regulated and 6 (7.32%) down-regulated out of the 82 quantified genes included in the dilated cardiomyopathy KEGG-constructed pathway [54]. The significantly regulated genes in this pathway were *Adcy1/4/5* (denylate cyclase 1/4/5); *Cacnb2* (calcium channel, voltage-dependent, beta 2 subunit); *Itga9/b1/b6* (integrin alpha 9/beta 1/beta 6); *Myh6/7* (myosin, heavy polypeptide heavy polypeptide 6, cardiac muscle, alpha/7, cardiac muscle, beta); *Myl2* (myosin, light polypeptide 2, regulatory, cardiac, slow); *Prkaca* (protein kinase, cAMP-dependent, catalytic, alpha); and *Tgfb3* (transforming growth factor, beta 3).

Figure S2 from the Supplementary Materials presents the positions of the 7 (8.86%) up-regulated and 6 (7.59%) down-regulated out of the 91 genes included in the hypertrophic cardiomyopathy KEGG-constructed pathway [55]. The HCM-regulated genes were *Cacnb2* (calcium channel, voltage-dependent, beta 2 subunit); *Edn1* (endothelin 1), *Itga9/b1/b6* (integrin alpha 9/beta 1/beta 6); *Myh6/7* (myosin, heavy polypeptide heavy polypeptide

6, cardiac muscle, alpha/7, cardiac muscle, beta); *Myl2* (myosin, light polypeptide 2, regulatory, cardiac, slow); *Tgfb3* (transforming growth factor, beta 3), *Tpm1* (tropomyosin 1, alpha); and *Tpm3* (tropomyosin 3, gamma).

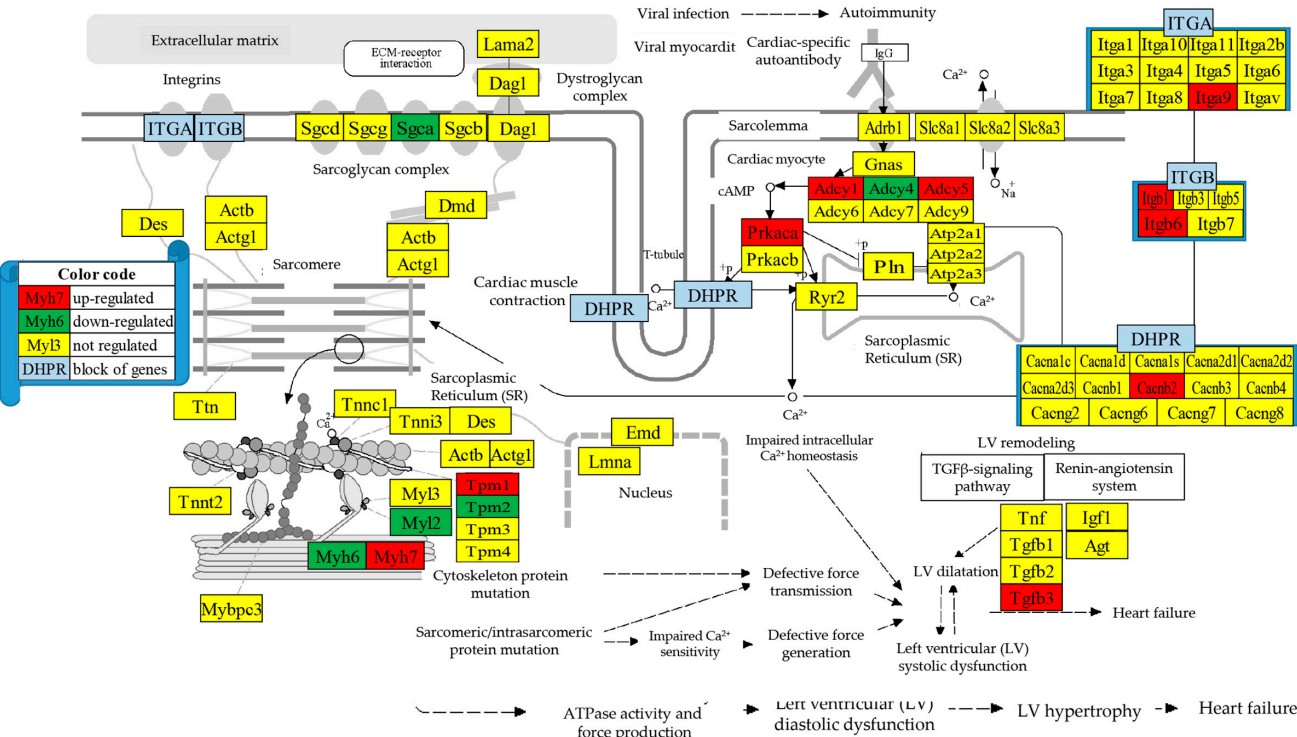

**Figure 5.** Regulated genes within the dilated cardiomyopathy KEGG-constructed pathway.

Figure 6 presents the positions of the 10 (11.76%) up-regulated and 3 (3.53%) down-regulated out of the 85 quantified genes included in the KEGG-constructed pathway of the parasitic Chagas disease [52]. Regulated genes: *Adcy1* (denylate cyclase 1), *Akt1/3* (thymoma viral proto-oncogene 1/3), *Casp8* (caspase 8), *Fadd* (Fas (TNFRSF6)-associated via death domain), *Ikbkg* (inhibitor of kappaB kinase gamma), *Irak1* (interleukin-1 receptor-associated kinase 1), *Mapk1/10* (mitogen-activated protein kinase 1/10), *Myd88* (myeloid differentiation primary response gene 88), *Ppp2r2a* (protein phosphatase 2, regulatory subunit B, alpha), *Tgfb3* (transforming growth factor, beta 3), *Tlr2* (toll-like receptor 2).

Figure 7 presents the positions of the regulated genes in the mitochondrial module of the diabetic cardiomyopathy KEGG-constructed pathway [53]. Regulated genes: *Atp5j* (ATP synthase, H+ transporting, mitochondrial F0 complex, subunit F), *Mpc2* (mitochondrial pyruvate carrier 2), *Ndufb11* (NADH: ubiquinone oxidoreductase subunit B11), *Ndufb4* (NADH: ubiquinone oxidoreductase subunit B4), *Ndufc1* (NADH: ubiquinone oxidoreductase subunit C1), *Uqcr10* (ubiquinol-cytochrome c reductase, complex III subunit X), *Uqcrh* (ubiquinol-cytochrome c reductase hinge protein).

### 3.10. Remodeling of the Gene Networks

We found that the transcriptomic networks correlating the genes within and between functional pathways strongly depend on the amount of salt in the diet. Figure 8 presents the ($p < 0.05$) significant synergistically/antagonistically/independently expressed genes within the dilated cardiomyopathy KEGG-constructed pathway (DIL, [54]); and the ($p < 0.05$) significant synergistic/antagonistic/independent coexpression of the CMC [51], OXP [35], and DCM [53] shared gene *Cox6b2* (cytochrome c oxidase subunit 6B2) with DIL genes in the two dietary conditions. Note that the low-salt diet coupled *Cox6b2* with DIL

genes through 18 significant synergisms (no antagonism or independence), while in the normal diet, it was only 1 antagonism (with *Cacng6*) and three significant independences, (with *Cacnb1*, *Cacng7*, *Cacng8*), with all four turned to significant synergisms by reducing the salt intake. We can also observe substantial remodeling within the DIL pathway. For instance, Atp2a2 is antagonistically coupled with four calcium channels (*Cacna1d*, *Cacna2d3*, *Cacnb3*, *Cagng2*) and two sodium/calcium exchangers (*Slc8a1*, *Slc8a2*) in the normal diet, but synergistically coupled with only one calcium channel (*Cacna1c*) in low-salt diet.

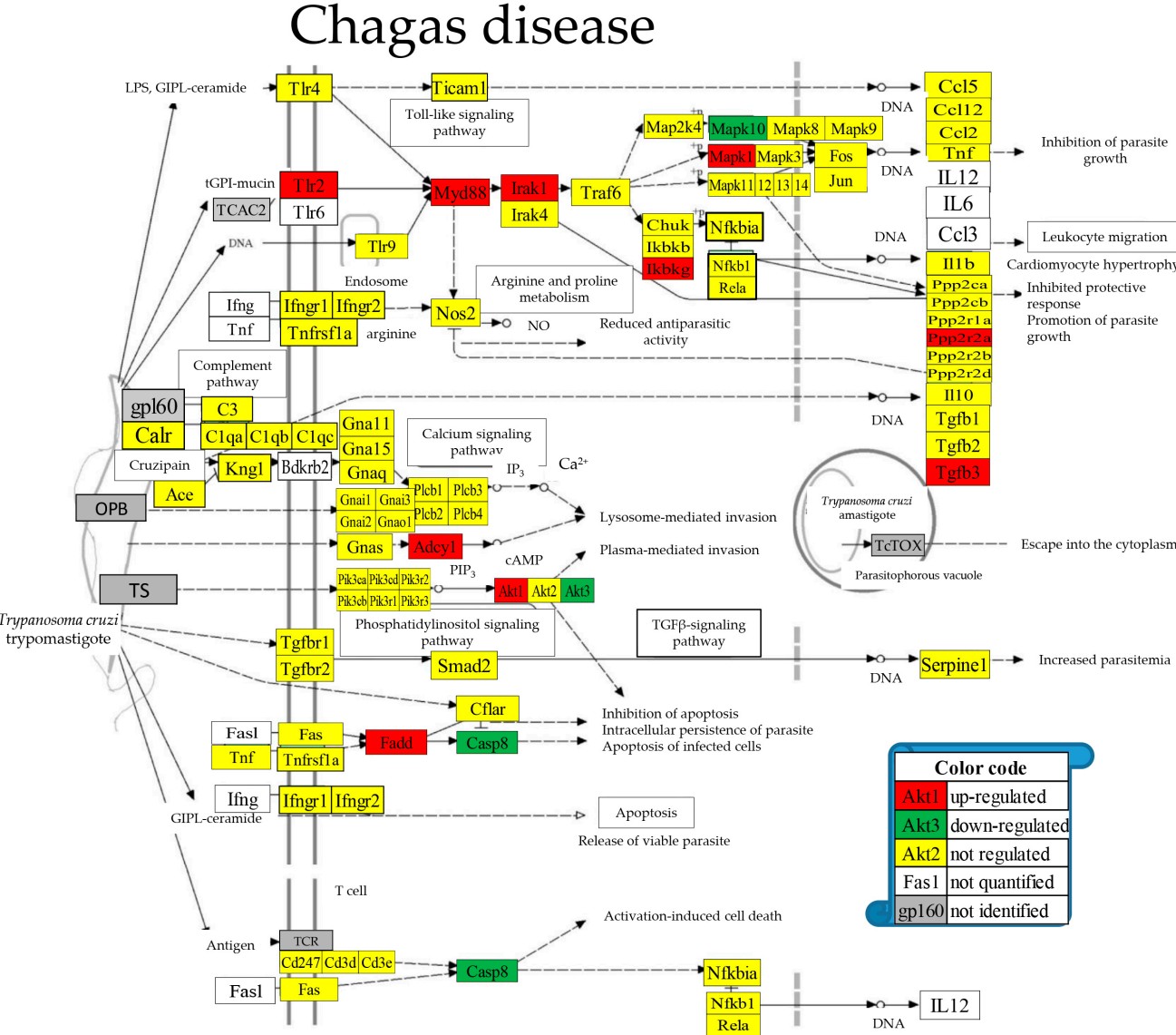

**Figure 6.** Regulated genes within the Chagas disease KEGG-constructed pathway [52].

Figure 9 presents the statistically ($p < 0.05$) significant synergistic/antagonistic/independent (red/green/yellow square) expression of several genes from the glycolysis/glucogenesis KEGG-constructed pathway (GLY, [33]), with those from cardiac muscle contraction (CMC, [51]) in the left ventricles of mice subjected to normal and low-salt diets. Of note is the almost compact expression coupling of the two pathways in the normal diet and the substantial decoupling in the low-salt diet. There are 302 (10.17%) synergistically, 246 (8.28%) antagonistically, and 54 (1.81%) independently expressed gene pairs among the 1485 distinct pairs that can be formed with the 55 GLY genes, yielding COORD = 16.63% in the normal diet. These numbers are reduced to 192 (6.47%) synergistic, 100 (3.67%) antagonistic, and 104 (3.50%) independent expressions in the low-salt diet,

making COORD = 6.33%. Among the 2775 distinct pairs that can be formed with CMC genes, 732 (13.19%) were synergistic, 404 (7.28%) antagonistic, and 138 (2.49%) independent in normal (COORD = 17.98%). The numbers of significant correlations became 514 (9.26%) synergistic, 68 (1.23%) antagonistic, and 168 (3.03%) independent (COORD = 7.46%) in the low-salt diet. The expression correlations between GLY and CMC genes (4125 distinct pairs) were also affected. A total of 496 (12.02%) synergisms, 311 (7.54%) antagonisms, and 94 (2.28%) independences in normal diet (COORD = 17.28%) became 309 (7.49%) synergisms, 110 (2.67%) antagonisms, and 127 (3.08%) independences (COORD = 7.08%) in low-salt diet.

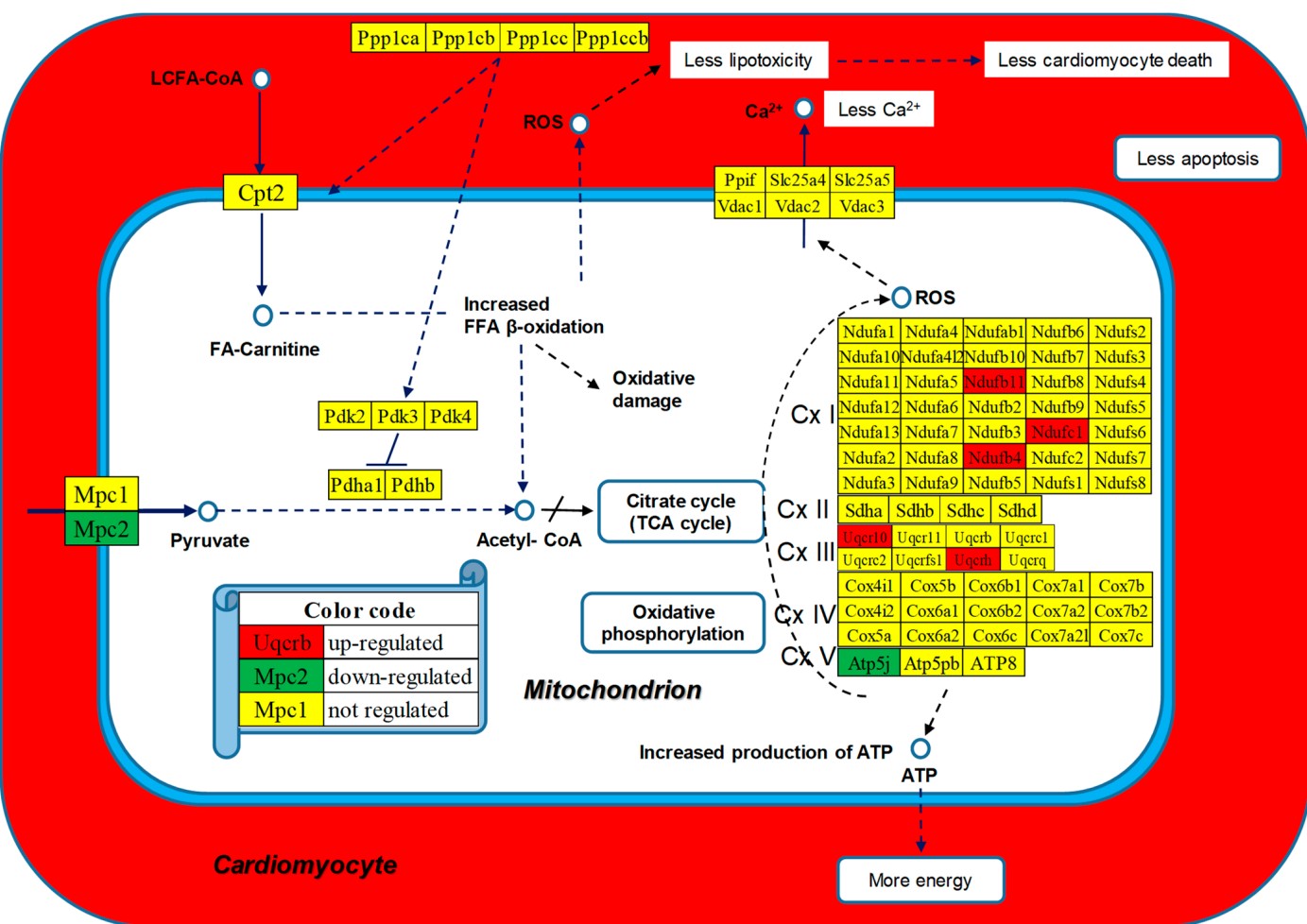

**Figure 7.** Regulated mitochondrial genes included in the diabetic cardiomyopathy KEGG-constructed pathway [53].

Figure 10 presents the statistically ($p < 0.05$) significant synergistic and antagonistic expression of several genes from the adrenergic signaling in cardiomyocytes KEGG-constructed pathway [50] with genes from the cardiac muscle contraction [51] and hypertrophic cardiomyopathy [55] pathways, in the left ventricle of mice fed with (A) normal diet and (B) low-salt diet. Of note again is the massive decoupling of the three pathways from 13.82% (ASC–CMC) and 10.50% (ASC–HCM) in normal salt to 2.91 (ASC–CMC) and 2.83% (ASC–HCM) in low salt, indicating a major remodeling of the interplay among these functional pathways.

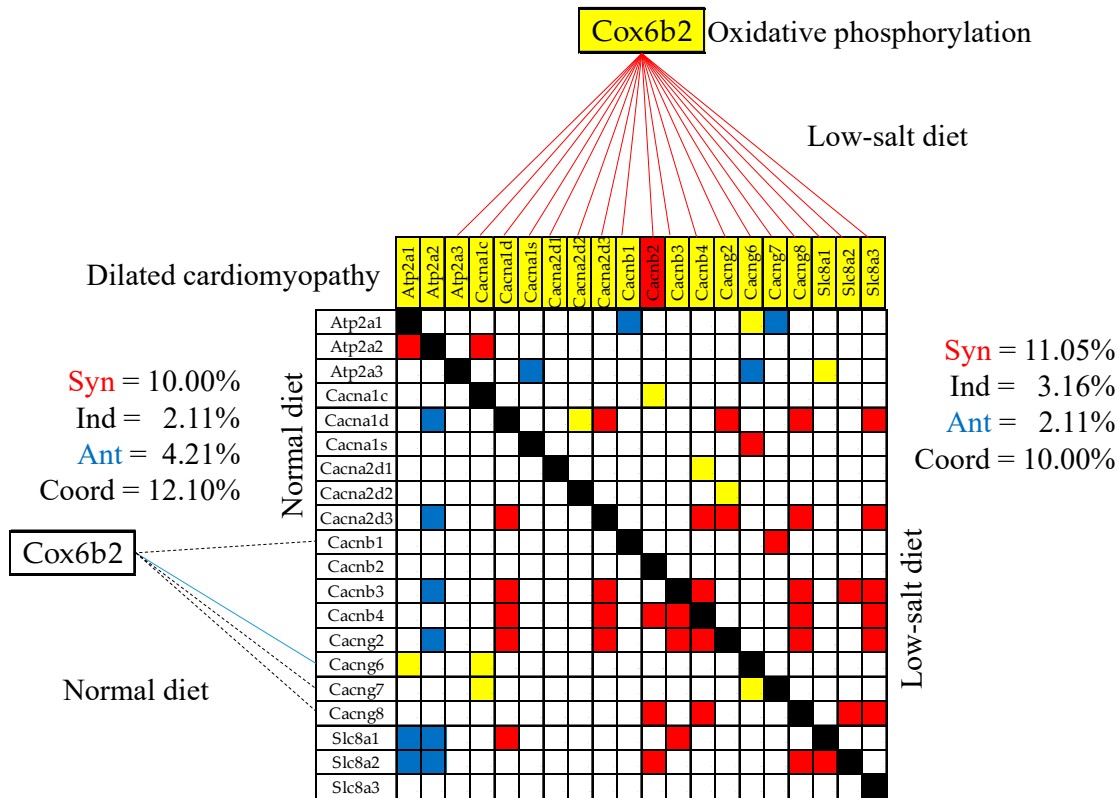

**Figure 8.** Statistically ($p < 0.05$) significant synergistically/antagonistically/independently expressed genes within the dilated cardiomyopathy (red/green/yellow squares) KEGG-constructed pathway; and the ($p < 0.05$) significant synergistic (continuous red line), antagonistic (continuous blue line), and independent (dashed black line) expression of *Cox6b2* (cytochrome c oxidase subunit 6B2) with genes involved in the dilated cardiomyopathy pathway in the left ventricles of mice fed with normal/low-salt diet. The red background of the *Cacnab2* gene symbol indicates significant up-regulation in low-salt with respect to the normal diet, while the yellow background of the other gene symbols indicates no significant regulation.

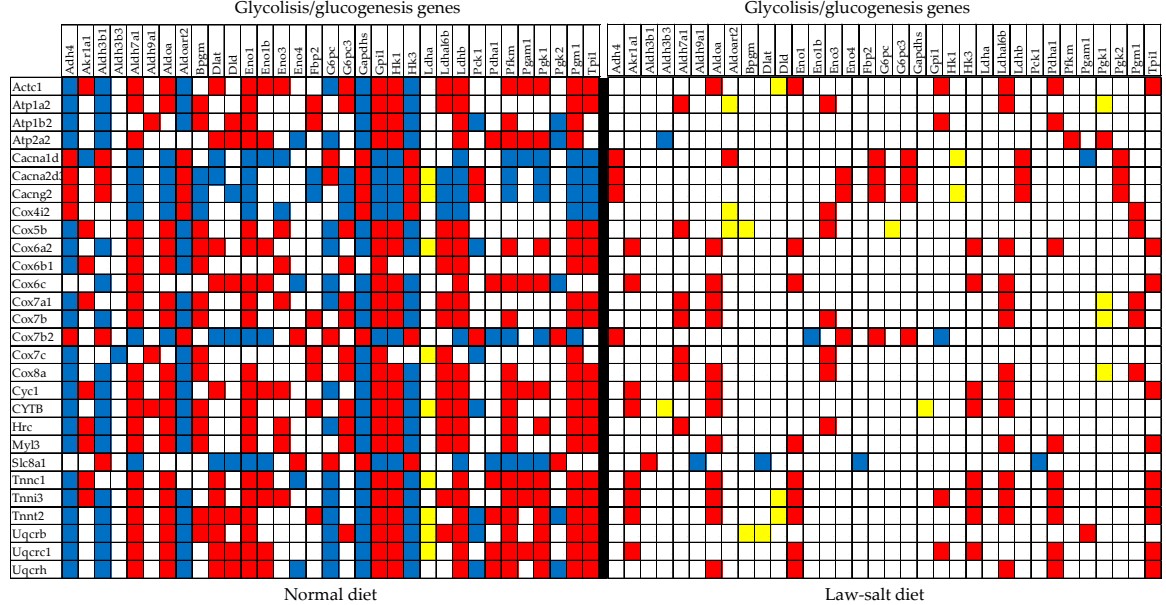

**Figure 9.** Statistically ($p < 0.05$) significant synergistic (red square), antagonistic (blue square), and independent (yellow square) expression of genes from the glycolysis/glucogenesis and cardiac muscle

contraction KEGG-constructed pathways in the normal and low-salt diets. Only the gene pairs with statistically significant synergistic, antagonistic, or independent expressions were represented. Of note is the almost compact expression coupling of the two pathways in the normal diet and the substantial decoupling in the low-salt diet.

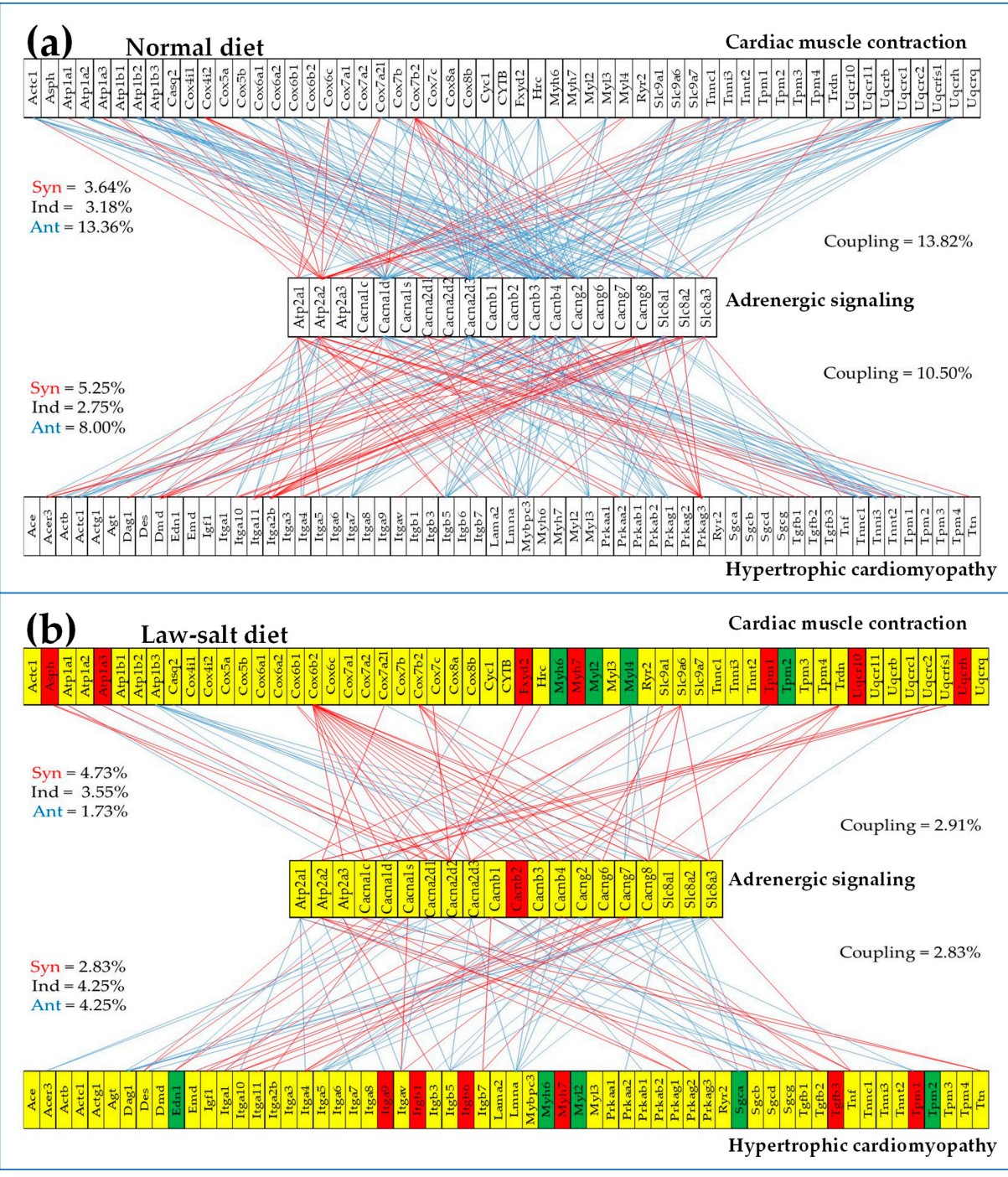

**Figure 10.** Statistically ($p < 0.05$) significant synergistic and antagonistic expression of several genes from the adrenergic signaling in cardiomyocytes KEGG-constructed pathway with genes from the cardiac muscle contraction and hypertrophic cardiomyopathy pathways, in the left ventricle of mice fed with (**a**) normal diet and (**b**) low-salt diet. Red/blue lines indicate synergistic/antagonistic expressions of the linked genes. The red/green gene symbol background in (**b**) indicates significant up-/down-regulation, while the yellow background indicates that the gene's expression was not significantly altered.

## 4. Discussion

Although sodium is just one out of numerous regulators of the heart function, there are still many unknowns about how a low-salt diet may reduce the risks of cardiac diseases. Gene expression profiling provides a very powerful way to decipher the molecular mechanisms.

We have analyzed expression data from a microarray experiment deposited in a publicly accessible database to determine the cardiogenomic effects of reducing the salt intake in the left heart ventricle of adult mice from the perspective of the Genomic Fabric Paradigm (GFP). Through characterizing each profiled gene by three types of independent measures, GFP provides the most theoretically possible comprehensive characterization of the transcriptome. As illustrated in Figure 1 for 55 glycolysis/glucogenesis genes, the relative expression variations (REVs) and the expression correlations (CORs) with each other gene are independent with respect to the average expression levels (AVEs). Thus, compared to the traditional gene expression analysis, GFP increased by almost four orders of magnitude the transcriptomic information collected from the analyzed microarray experiment, adding very important, yet still neglected, transcriptomic measures.

While the universally-used AVE is good for identifying what gene was significantly up-/down-regulated when comparing an experimental condition with the corresponding control (pending the appropriate cut-off criteria), it is REV that provides a measure of the strength of the homeostatic control of transcript abundance. Thus, the high REV (101.47) of *Pck2* indicates a very relaxed control of the expression level of this gene, making it a good vector of adaptation to altered external conditions, including hypoxia [75].

In turn, COR analysis determines the most probable gene networking in functional pathways. It is based on the Principle of Transcriptomic Stoichiometry [76,77] that requires the networked genes to be coordinately expressed to ensure the efficiency of the functional pathway. Among much other interesting information, Figure 1 premieres the glycolysis/glucogenesis expression coordination partners of *Slc8a1*, a key gene for calcium homeostasis whose inactivation limits the damages caused by myocardial infarction [78] and the dependence on diet of the partnership.

The primary independent characteristics allowed us to define some important derived characteristics to deepen the understanding of heart genomics. For instance, through the relative expression control (REC), we obtained insights about the cell priorities in ensuring the right amounts of transcripts. For now, there is no information in PubMed, and also we do not have any hypothesis of why *Aldh3a2* is by far the most protected member of the aldehyde dehydrogenase family in a normal diet and what caused its substantial fall from the cell's interest in a low-salt diet. However, this gene, and also the other highly protected GLY gene, *Galm*, deserve further investigation for their roles in normal heart physiology, beyond their direct involvement in carbohydrate metabolism.

The high GCH (33.64) of the CMC gene *Cox4i2* in the normal heart looks deserved given how essential the encoded protein is for acute pulmonary oxygen sensing [79]. The reduction in GCH to 2.67 in a low-salt diet might be interpreted as better protection of the heart in this diet against life-threatening hypoxemia.

As illustrated in Table 1, our composite criterion with absolute fold-change cut-off calculated for every gene to identify the significantly regulated genes proved efficient in eliminating numerous false positive hits and adding several missed genes caused by the fixed $1.5\times$ cut-off. As well, it justified the addition of other genes whose significant regulation would have been neglected by the traditional analysis. There are several important genes for heart physiology whose significant up-regulation was revealed by our algorithm (like *Myd88* (myeloid differentiation primary response gene 88): an important mediator of the inflammatory signaling carried by the toll-like and Il-1 families of receptors [80]). Other important up-regulated genes were *Fxyd2* (FXYD domain-containing ion transport regulator 2), an important regulator of the $Na^+$ transport [81], and *Itgb6* (myo-inositol 1-phosphate synthase A1), involved in resynchronization following heart failure [82]. From the identified down-regulated genes, of note are *Gsk3b* (glycogen synthase kinase-3β), a critical regulator of cell proliferation and differentiation [83]; *Chat* (choline acetyltrans-

ferase), related to the ventricular remodeling in type 1 diabetes [84]; and *Cmpk2* (cytidine monophosphate), involved in inflammatory diseases [85].

We prefer to use WIR (illustrated in Figure 3b) as a more adequate measure to characterize the expression regulation of individual genes and their contribution to the overall contributions to transcriptomic alteration. From this perspective, the largest positive contributions were delivered by *Rrp36* (ribosomal RNA processing 36 homologs) and *Uqcrh* (ubiquinol-cytochrome c reductase hinge protein, WIR = 203). While *Uqcrh* is directly involved in the CMC [51], OXP [35], and DIA [54] KEGG-constructed pathways, *Rrp36* is one of the major cellular activity mobilizing genes [86] and its up-regulation indicates the benefits of reducing salt intake. The encoded protein of the most up-regulated gene, *Prg4* (proteoglycan 4 (megakaryocyte stimulating factor, articular superficial zone protein), x = 196), was associated with the slope of the body mass index [87]. The largest negative contributions were provided by *Ccdc157* (coiled-coil domain containing 157, WIR = −1472, x = 69.85) and *Cdca8* (cell division cycle associated 8, WIR = −556, x = −56.33). *Ccdc157* was identified as important in the protein and trafficking pathways [88].

The WPR analysis (Table 2) indicated CMC, OXP, and the mitochondrial module of DIA as the most improved among the selected pathways in the experimental diet through the up-regulated myosines, tropomyosines, and genes of respiratory chain complexes I and III. It is interesting to note the large contributions of the respiratory genes from Complex I (*Ndufb4*, WIR = 95.91; *Ndufc1*, WIR = 58.83), and those from Complex III (*Uqcr10*, WIR = 177.85 and *Uqcrh*, WIR = 202.92), that might have increased the production of ATP. By contrast, the negative contribution of the pyruvate transporter *Mcp2* (WIR = −76.16) may finally lead to the reduction in the reactive oxygen species, increasing the viability of the hosting cardiomyocyte (Figure 7).

Analysis of the regulation of expression control (illustrated in Figure 3c for several purine metabolism genes) provides additional, non-redundant information about the LSD transcriptomic effects on the heart's left ventricle. Of all 19,605 quantified genes, the largest increase in ΔREC in LSD was exhibited by *Usp31* (ΔREC = 2411%), a potential biomarker [89] for clear cell renal cell carcinoma [90] and *Syt11* (ΔREC = 1517%), known for its role in atrial fibrillation [72]. In contrast, *Mcph1* (microcephaly, primary autosomal recessive 1, ΔREC = −3515%), involved in determining the mitral valve diameter [71] and DNA-damage signaling and repair [91], and *Aldh3a2* (ΔREC = −1559%) had the largest reduction in the expression control.

LSD resulted in many more up-regulated than down-regulated genes within the metabolic (Table 3, up/down ratio = 97/66 = 1.47) and signaling (Table 4, up/down ratio = 607/350 = 1.73) pathways, indicating increased efficiency of metabolism and signaling. Although none of the quantified alpha (*Adra1a*, *Adra1b*, *Adra1*) and beta (*Adrb1*, *Adrb2*) adrenergic receptors were regulated (Figure 4), the inward sodium transporters *Scn1b* and *Scn5a* were over-expressed, presumably to compensate for the low sodium level: this might be relevant in the treatment of Brugada syndrome [92]. Also up-regulated was the $Na^+$-$K^+$ exchanger *Atp1a3* whose mutations are related to several neurological and cardiovascular diseases [93].

We found interesting LSD consequences on the pathways of several cardiomyopathies that should be considered when deciding about the treatment options. For instance, the up-regulation of the integrins *Itga9*, *Itgb1*, *and Itgb6* (Figure 6), important membrane adhesion receptors involved in both inside-out and outside-in signaling of cardiomyocytes, might have direct consequences on the therapeutic efficiency of their inhibitors [94]. The down-regulation of *Casp8* (Figure 7) reduced the apoptosis risk [95] in cardiomyocytes elevated by the up-regulation of *Fadd* [96] in Chagas disease [97] following infection with *Trypanosoma cruzi* [98].

While the LSD effects on the gene and protein expression have been reported in numerous studies (e.g., [99–101]), this is the first time, to our knowledge, that remodeling of the gene transcriptomic networks is reported. As shown in Figures 8–10, the LSD-induced remodeling affects the gene expression intercoordination both within functional

pathways and between interacting pathways. Interestingly, LSD significantly reduced the coordination degrees within CMC (from 12.10% to 10.00%, Figure 8) and GLY (from 16.63% to 6.33%) pathways. The expression coordination was also significantly reduced between GLY and CMC (from 17.28% to 7.49%, Figure 9), between ASC and CMC (from 13.82% to 2.91%), and between ASC and HCM (from 10.50% to 2.83%, Figure 10). This substantial decoupling within, as well as among, functional pathways most likely increases the flexibility and adaptability of the heart's physiology to external stimuli.

## 5. Conclusions

Using the mathematically advanced GFP algorithms, the study revealed for the first time that, in addition to regulating expression of numerous genes, LSD affects the homeostatic control of the transcripts' abundances and remodels the transcriptomic networks linking genes within and between functional pathways.

The study was limited to male mice because female transcriptome is strongly dependent on the estrogen level [102]. Therefore, our future research will extend the left ventricle gene expression profiling to female mice synchronized for each of the four phases of the estrogen cycle, to see how the female sex benefits from the low-salt diet. Moreover, owing to the integration of the cardiovascular system in the general physiology, an ideal research project would simultaneously investigate the LSD-induced genomic changes in the metabolism, and intercellular signaling also present in the kidneys, liver, pancreas, stomach, and other related organs.

**Supplementary Materials:** The following supporting information can be downloaded at: https://www.mdpi.com/article/10.3390/cimb46030150/s1, Figure S1: Gene Commanding Heights (GCH) within the KEGG-constructed CMC (cardiac muscle contraction) pathway [51]. Note the reduction in the GCH scores for most CMC genes in LSD. Figure S2: Regulated genes in the hypertrophic cardiomyopathy KEGG-constructed pathway. Regulated genes: *Cacnb2* (calcium channel, voltage-dependent, beta 2 subunit); *Edn1* (endothelin 1), *Itga9* (integrin alpha 9); *Itgb1* (integrin beta 1); *Itgb6* (integrin beta 6); *Myh6/7* (myosin, heavy polypeptide 6, cardiac muscle, alpha/7, cardiac muscle, beta); *Myl2/4* (myosin, light polypeptide 2/4); *Sgca* (sarcoglycan, alpha (dystrophin-associated glycoprotein)); *Tgfb3* (transforming growth factor, beta 3); *Tpm1/2* (tropomyosin 1 alpha/2 beta).

**Author Contributions:** Conceptualization, D.A.I.; methodology, D.A.I. and S.I.; software, D.A.I.; validation, D.A.I. and S.I.; formal analysis, S.I.; investigation, D.A.I.; resources, D.A.I.; data curation, S.I.; writing—original draft preparation, D.A.I.; writing—review and editing, H.A.; visualization, H.A. and D.A.I.; supervision, D.A.I.; project administration, D.A.I.; funding acquisition, D.A.I. All authors have read and agreed to the published version of the manuscript.

**Funding:** This research received no external funding.

**Institutional Review Board Statement:** Not applicable (expression data downloaded from a publicly accessible database).

**Data Availability Statement:** Experimental details and raw data are available online at https://www.ncbi.nlm.nih.gov/geo/query/acc.cgi?acc=GSE72561 (accessed on 6 February 2024)

**Acknowledgments:** We acknowledge the constant support of Denis Daniels, Director of the PVAMU Undergraduate Medical Academy.

**Conflicts of Interest:** The authors declare no conflicts of interest.

## Appendix A  Independent Primary Expression Characteristics of Individual Gene and Functional Pathways

1. (Normalized) Average Expression Level (*AVE*) of gene *i* in condition $c = N, L$, probed redundantly by $R_i$ microarray spots was normalized to the median gene expression in that condition:

$$AVE_i^{(c)} \equiv \frac{\frac{1}{4}\sum_{k=1}^{4}\left(\frac{1}{R_i}\sum_{r_i=1}^{R_i} a_{i;k;r_i}^{(c)}\right)}{\left\langle \frac{1}{4}\sum_{k=1}^{4}\left(\frac{1}{R_j}\sum_{r_j=1}^{R_j} a_{j;k;r_j}^{(c)}\right)\right\rangle\Big|_{all\ j}}, \tag{A1}$$

where:

$\langle B_j \rangle|_{allj}$ = median B over all quantified genes

$a_{i;k;r_i}^{(c)}$ = background subtracted fluorescence of spot $r_i$ probing gene $i$ in replica $k$

$R_i$ = the number of microarray spots redundantly probing transcript $i$.

*AVE* of individual genes can be averaged within a particular functional pathway $\Gamma$:

$$AVE_\Gamma^{(c)} \equiv \frac{1}{\{\Gamma\}} \sum_{i\epsilon\Gamma} AVE_i^{(c)} \text{ , } \{\Gamma\} \equiv \text{number of genes in pathway } \Gamma \tag{A2}$$

2. Relative Expression Variation (*REV*) is defined as the mid-interval chi-square estimate with probability $\alpha = 0.05$ of the coefficient of variation in gene $i$ in condition $c = N$, $L$, probed redundantly by $R_i$ microarray spots in all four biological replicas:

$$REV_i^{(c)} \equiv \frac{1}{2}\left(\sqrt{\frac{4R_i-1}{\chi^2(4R_i-1;0.975)}} + \sqrt{\frac{4R_i-1}{\chi^2(R_i-1;0.025)}}\right)\sqrt{\frac{1}{R_i}\sum_{r_i=1}^{R_i}\left(\frac{s_{i;r_i}^{(c)}}{{}_{i;r_i}^{(c)}}\right)^2} \times 100\% \tag{A3}$$

where:

$\mu_{i;r_i}^{(c)} \equiv \frac{1}{4}\sum_{k=1}^4 a_{i;k,r_i}^{(c)}, s_{i;r_i}^{(c)} \equiv \sqrt{\frac{\sum_{k=1}^4\left(a_{i;k,r_i}^{(c)}-\mu_{i;r_i}^{(c)}\right)^2}{3}}$,

$\chi^2$ = chi-square test statistic with 4 $R_i$ degrees of freedom and probability $\alpha = 0.05$

*REV* of individual genes can be averaged within a particular functional pathway $\Gamma$:

$$REV_\Gamma^{(c)} \equiv \frac{1}{\{\Gamma\}}\sum_{i\epsilon\Gamma} REV_i^{(c)}, \{\Gamma\} \equiv \text{number of genes in pathway } \Gamma \tag{A4}$$

3. Expression correlation (*COR*) of gene $i$ with gene $j$ in condition $c = N$, $L$, probed redundantly by $R_i$ and $R_j$ microarray spots in all four biological replicas:

$$COR_{i,j}^{(c)} \equiv \frac{\sum_{k=1}^4\sum_{r_i=1}^{R_i}\sum_{r_j=1}^{R_j}\left(a_{i;k,r_i}^{(c)} - {}_{i;r_i}^{(c)}\right)\left(a_{j;k,j}^{(c)} - {}_{j;j}^{(c)}\right)}{\sqrt{\left(\sum_{k=1}^4\sum_{r_j=1}^{R_j}\left(a_{j;k,r_j}^{(c)} - {}_{j;r_j}^{(c)}\right)^2\right)\left(\sum_{k=1}^4\sum_{r_i=1}^{R_i}\left(a_{i;k,r_i}^{(c)} - {}_{i;r_i}^{(c)}\right)^2\right)}} \tag{A5}$$

One may note that COR is, actually, the pair-wise Pearson's coefficient of correlation between two sets of data.

*COR* of individual genes can be averaged within a particular functional pathway $\Gamma$:

$$COR_\Gamma^{(c)} \equiv \frac{1}{\{\Gamma\}}\sum_{i\epsilon\Gamma} COR_i^{(c)}, \{\Gamma\} \equiv number of genes in pathway \Gamma \tag{A6}$$

## Appendix B Derived Characteristics of Individual Genes and Their Averages Over Functional Pathways

1. Relative Expression Control:

$$REC_i^{(c)} \equiv \frac{\langle REV_j^{(c)}\rangle|_{all \, j}}{REV_i^{(c)}} \tag{A7}$$

*REC* of individual genes can be averaged within a particular functional pathway $\Gamma$:

$$REC_\Gamma^{(c)} \equiv \frac{1}{\{\Gamma\}}\sum_{i\epsilon\Gamma} REC_i^{(c)}, \{\Gamma\} \equiv number of genes in pathway \Gamma \tag{A8}$$

2. Coordination degree of individual genes:

$$COORD_i^{(c)} \equiv SYN_i^{(c)} + ANT_i^{(c)} - IND_i^{(c)} \tag{A9}$$

*SYN*, *ANT*, and *IND* are the percentages of genes forming with gene *i* ($p < 0.05$) statistically significant synergistic, antagonistic, or independent expressed pairs across the biological replicas. The analysis can cover the entire transcriptome or be restricted to a particular functional pathway.

*COORD* of individual genes can be averaged within a particular functional pathway $\Gamma$:

$$COORD_\Gamma^{(c)} \equiv \frac{1}{\{\Gamma\}} \sum_{i \epsilon \Gamma} COORD_i^{(c)}, \ \{\Gamma\} \equiv \text{number of genes in pathway } \Gamma \tag{A10}$$

*COORD* of individual genes can be averaged between two functional pathways $\Gamma$ and $\Theta$:

$$COORD_{\Gamma,\Theta}^{(c)} \equiv \frac{1}{\{\Gamma\}\{\Theta\}} \sum_{\substack{i \epsilon \Gamma, j \in \Theta \\ j \neq i}} COORD_{i,j}^{(c)}, \ \{\Gamma\}, \{\Theta\} \equiv \text{numbers of genes in } \Gamma \text{ and } \Theta \text{ pathways} \tag{A11}$$

3. Gene Commanding Height of individual genes:

$$GCH_i^{(c)} \equiv REC_i^{(c)} exp\left(\frac{4}{N} \sum_{j=1}^{N} \left(COR_{i,j}^{(c)}\right)^2\right) \tag{A12}$$

*GCH* of individual genes can be averaged within a particular functional pathway $\Gamma$:

$$GCH_\Gamma^{(c)} \equiv \frac{1}{\{\Gamma\}} \sum_{i \epsilon \Gamma} GCH_i^{(c)}, \ \{\Gamma\} \equiv \text{number of genes in pathway } \Gamma \tag{A13}$$

**Appendix C  Measures of Transcriptomic Regulation**

1. Statistically significant regulation of the average expression level:

$$\left|x_i^{(L \to N)}\right| > CUT_i^{(L \to N)} \equiv 1 + \sqrt{2\left(\left(\frac{REV_i^{(N)}}{100}\right)^2 + \left(\frac{REV_i^{(L)}}{100}\right)^2\right)} \ \& \ p_i^{(L \to N)} < 0.05$$

$$x_i^{(L \to N)} = \begin{cases} \frac{AVE_i^{(L)}}{AVE_i^{(N)}} & if: \ AVE_i^{(L)} > AVE_i^{(N)} \\ -\frac{AVE_i^{(N)}}{AVE_i^{(L)}} & if: \ AVE_i^{(L)} \leq AVE_i^{(N)} \end{cases} \tag{A14}$$

2. Weighted Individual (gene) Regulation (*WIR*)

$$WIR_i^{(L \to N)} \equiv AVE_i^{(N)} \frac{x_i^{(L \to N)}}{Abs\left(x_i^{(L \to N)}\right)} \left(Abs\left(x_i^{(L \to N)}\right) - 1\right) \left(1 - p_i^{(L \to N)}\right) \tag{A15}$$

*WIR* of individual genes can be averaged within a particular functional pathway $\Gamma$:

$$WPR_\Gamma^{(L \to N)} \equiv \sqrt{\frac{\sum_{i \epsilon \Gamma} \left(WIR_i^{(L \to N)}\right)^2}{\{\Gamma\}}}, \ \{\Gamma\} \equiv \text{number of genes in pathway } \Gamma \tag{A16}$$

3. Regulation of the expression control of individual genes:

$$\Delta REC_i^{(L \to N)} = \left(\frac{1}{REV_i^{(L)}} - \frac{1}{REV_i^{(N)}}\right) \times 100\% \tag{A17}$$

$\Delta REC$ of individual genes can be averaged within a particular functional pathway $\Gamma$:

$$\Delta REC_\Gamma^{(L \to N)} = \frac{1}{\{\Gamma\}} \sum_{i \epsilon \Gamma} \left( \frac{1}{REV_i^{(L)}} - \frac{1}{REV_i^{(N)}} \right) \times 100\% \qquad (A18)$$

4. Regulation of the expression coordination of individual genes:

$$\Delta COR_{i,\Gamma}^{(L \to N)} = \frac{\sum_{j \in \Gamma}\left( COR_{i,j}^{(L)} - COR_{i,j}^{(N)} \right)}{Abs\left( \sum_{j \in \Gamma}\left( COR_{i,j}^{(L)} - COR_{i,j}^{(N)} \right) \right)} \sqrt{\frac{\sum_{j \epsilon \Gamma}\left( COR_{i,j}^{(L)} - COR_{i,j}^{(N)} \right)^2}{\{\Gamma\}}} \qquad (A19)$$

5. Regulation of the coordination degree within a functional pathway:

$$\Delta\Delta COORD_i^{(L \to N)} \equiv \sum_{i \epsilon \Gamma}\left( COORD_i^{(L)} - COORD_i^{(N)} \right)$$

$$COR_{\Gamma \to \Gamma}^{(L \to N)} = \frac{\sum_{i,j \in \Gamma}\left( COR_{i,j}^{(L)} - COR_{i,j}^{(N)} \right)}{Abs\left( \sum_{i,j \in \Gamma}\left( COR_{i,j}^{(L)} - COR_{i,j}^{(N)} \right) \right)} \sqrt{\frac{\sum_{i,j \epsilon \Gamma}\left( COR_{i,j}^{(L)} - COR_{i,j}^{(N)} \right)^2}{\{\Gamma\}\{\Gamma\}}} \qquad (A20)$$

6. Overall regulation of the expression coordination between functional pathways:

$$COR_{\Gamma \to \Theta}^{(L \to N)} = \frac{\sum_{i \in \Gamma, j \epsilon \Theta}\left( COR_{i,j}^{(L)} - COR_{i,j}^{(N)} \right)}{Abs\left( \sum_{i \in \Gamma, j \epsilon \Theta}\left( COR_{i,j}^{(L)} - COR_{i,j}^{(N)} \right) \right)} \sqrt{\frac{\sum_{i \epsilon \Gamma, , j \epsilon \Theta}\left( COR_{i,j}^{(L)} - COR_{i,j}^{(N)} \right)^2}{\{\Gamma\}\{\Theta\}}} \qquad (A21)$$

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
