# Peer review of "Low-Salt Diet Regulates the Metabolic and Signal Transduction Genomic Fabrics, and Remodels the Cardiac Normal and Chronic Pathological Pathways"

_cimb, doi:10.3390/cimb46030150_

Round 1
Reviewer 1 Report
Comments and Suggestions for Authors
Dear Authors,
I wish to congratulate you with the submitted manuscript.
It's very rare when I can recommend the manuscript to be accepted without further suggestions.
kinds
R
Author Response
Thank you so much for the kind appreciation of our work.
Reviewer 2 Report
Comments and Suggestions for Authors
This is a straightforward study with an unsurprising result in that chronic maintenance of a murine study population on a low sodium diet relative to a control population would be expected to alter the metabolism and transcriptome of multiple organs, although the focus was on the left ventricular myocardium. Inclusion of data from other organ would increase the significance overall, as well as the inclusion of data collected from mice maintained on a hypernatremic diet. The most significant finding of the study was that a low sodium diet appears to be beneficial in the context of various cardiomyopathies.
Overall, the study design is sound and there are no issues with the analysis or presentation of the data. The manuscript does need to be edit to fix various typos and grammatical errors.
Comments on the Quality of English LanguageSee above
Author Response
Thank you so much for your kind appreciation. We have corrected the typos and grammar errors.
Reviewer 3 Report
Comments and Suggestions for Authors
I carefully read the manuscript prepared by Dumitru Iacobas et al., and, in my opinion, it is a complex and well-written study based on the advanced mathematic algorithm in a significant area of research - low-salt diet impact into cardiac pathophysiology. Thus, I recommend this manuscript as a suitable research article for publication in the “Current Issues in Molecular Biology”, with a minor comment: the authors should address and expand, maybe at the end of the Discussion section, the limits of this study results, based on the complexity of the factors that control the cardiac function, not limited to salts diet.
In addition, please carefully revise the references according to the Current Issues in Molecular Biology journal recommendation.
Author Response
Thank you so much for the nice appreciation of our manuscript. We responded to your suggestions by mentioning that sodium intake is just one out of many factors regulating the heart physiology and also that the study should be extended to other organs like the kidney, liver, stomach, and pancreas.